# LET PHYSICS GUIDE YOUR PROTEIN FLOWS: TOPOLOGY-AWARE UNFOLDING AND GENERATION

## ABSTRACT

Protein structure prediction and folding are fundamental to understanding biology, with recent deep learning advances reshaping the field. Diffusion-based generative models have revolutionized protein design, enabling the creation of novel proteins. However, these methods often neglect the intrinsic physical realism of proteins, driven by noising dynamics that lack grounding in physical principles. To address this, we first introduce a physically motivated non-linear noising process, grounded in *classical physics*, that unfolds proteins into secondary structures (e.g., $\alpha$-helices, linear $\beta$-sheets) while preserving topological integrity—maintaining bonds and preventing collisions. We then integrate this process with the flow-matching paradigm on SE(3) to model the invariant distribution of protein backbones with high fidelity, incorporating sequence information to enable sequence-conditioned folding and expand the generative capabilities of our model. Experimental results demonstrate that the proposed method achieves state-of-the-art performance in unconditional protein generation, producing more designable and novel protein structures while accurately folding monomer sequences into precise protein conformations.

## 1 INTRODUCTION

Proteins participate in most cellular processes and consist of up to half of all biomass on earth (Schröder, 2017). Understanding their three-dimensional structures is essential to understanding life and addressing global health challenges (Cao et al., 2020; Silva et al., 2019).

In protein *folding*, the physical process of transforming a linear amino acid chain into a functional macromolecule is studied (Dill and MacCallum, 2012; Frauenfelder, 2010), with several deep learning breakthroughs in recent years (AlQuraishi, 2019; Jumper et al., 2021; Abramson et al., 2024). On the other hand, *generative* protein models aim to learn the distribution $p(\mathbf{x})$ of physically realistic, folded protein structures $\mathbf{x}$ (Watson et al., 2023; Ingraham et al., 2023; Geffner et al., 2025). Many of these approaches adopt the highly successful diffusion (Song et al., 2021b; Ho et al., 2020) and flow-model (Grathwohl et al., 2018; Lipman et al., 2023) paradigms, where a noise process perturbs proteins into a 'soup' of disconnected residues and a denoiser network learns to reconstruct the folded structure.

Recent advances have extended these models to the space of rotations and translations of protein backbones, yielding SE(3)-invariant generative models (Yim et al., 2023b;a; Bose et al., 2024; Huguet et al., 2024). These models rely on linear forward noising processes for computational simplicity. However, while effective, they overlook the physical realism of proteins and compromise structural integrity. This inconsistency propagates into the reverse process, making models prone to steric clashes and violations. We therefore study the following question:

> *How can we leverage physics to devise a physically motivated*
> *forward noising process for generative models in de novo protein design?*

**Contributions.** Toward that end, we introduce a physically motivated noising process derived from Hamiltonian dynamics (Hamilton, 1834) which incorporates simpler yet powerful inductive biases that ensure structural integrity and avoid collisions. We then learn to reverse this process using flow matching on SE(3), further incorporating sequence information to unify structure generation and folding. In particular, our contributions are,

Table 1: **Overview of deep learning methods for protein structure generation.** [†] indicates the methods that only penalize the model to be bond-persistent at the final steps of sampling. We use the following abbreviations: DDPM (Denoising Diffusion Probabilistic Models), CFM (Conditional Flow Matching), OT (Optimal Transport), and DSM (Denoising Score Matching).

| Method | Model | Backbone preserving | Collision free | Sequence Augmented | Coordinate system | Citation |
|---|---|---|---|---|---|---|
| RFDiffusion | DDPM | ✗ | ✗ | ✗ | Cartesian | Watson et al. (2023) |
| Chroma | DDPM | ✗ | ✗ | ✗ | Cartesian | Ingraham et al. (2023) |
| FrameFlow | CFM | ✓[(†)] | ✗ | ✗ | Frames | Yim et al. (2023a) |
| FoldFlow | CFM/OT | ✓[(†)] | ✗ | ✗ | Frames | Bose et al. (2024) |
| FoldFlow-2 | CFM/OT | ✓[(†)] | ✗ | ✓ | Frames | Huguet et al. (2024) |
| MultiFlow | CFM | ✗ | ✗ | ✓ | Frames | Campbell et al. (2024) |
| FrameDiff | DSM | ✓[(†)] | ✗ | ✗ | Frames | Yim et al. (2023b) |
| FoldingDiff | DDPM | ✓ | ✗ | ✗ | Angular | Wu et al. (2024) |
| **PhysFlow** | CFM | ✓ | ✓ | ✓ | Frames + Angular | this work |

- We propose a physics-inspired non-linear noising process that unfolds proteins into secondary structures while preserving structural integrity and avoiding collisions.

- We propose **PhysFlow**, which integrate this process with the flow-matching paradigm on $\mathrm{SE}(3)$ to model protein backbones, and incorporate sequence information to enhance generative capabilities.

- Empirically, PhysFlow achieves state-of-the-art performance in unconditional protein backbone generation and sequence-conditioned monomer folding.

## 2 BACKGROUND ON PROTEIN GENERATIVE MODELS

**Protein Backbone Parametrization**  We adopt the backbone frame parameterization introduced in AlphaFold2 (Jumper et al., 2021). For a protein of length $N$, the structure is represented by $N$ frames, each of which is SE(3)-equivariant. A frame is defined relative to fixed reference coordinates (idealized values of Alanine) $\mathrm{N}^*, \mathrm{C}_\alpha^*, \mathrm{C}^*, \mathrm{O}^* \in \mathbb{R}^3$, with $\mathrm{C}_\alpha^* = (0,0,0)$ being the origin, assuming idealized bond angles and lengths (Engh and Huber, 2006). The coordinates of residue $i$ are obtained by applying a rigid-body transformation $T_i = (\mathbf{r}_i, \mathbf{x}_i) \in \mathrm{SE}(3)$ to the reference coordinates, i.e.,

$$[\mathrm{N}_i, (\mathrm{C}_\alpha)_i, \mathrm{C}_i] = T_i \cdot [\mathrm{N}^*, \mathrm{C}_\alpha^*, \mathrm{C}^*] \tag{1}$$

where $r_i \in \mathrm{SO}(3)$ denotes the rotation matrix and $x_i \in \mathbb{R}^3$ denotes the translation. Collectively, the set of transformations $\mathbf{T} = [T_1, \ldots, T_N] \in \mathrm{SE}(3)^N$ represent the protein.

In addition, we also use an angular representation, denoted by $\mathbf{z} = (\mathbf{z}_1, \ldots, \mathbf{z}_N)$, where each residue $\mathbf{z}_i = (\phi, \psi, \omega, \theta_1, \theta_2, \theta_3) \in [-\pi, \pi]^6$ is represented using the three dihedral $(\phi, \psi, \omega)$ and bond $(\theta_1, \theta_2, \theta_3)$ angles of the backbone. Cartesian coordinates $\mathbf{x}$ can then be reconstructed from these angles via the mp-NeRF algorithm (Parsons et al., 2005; Alcaide et al., 2022) in a differentiable way, i.e. $\mathbf{x} = \mathrm{nerf}(\mathbf{z})$. See Appendix C for a primer on the coordinate systems.

**Flow models**  Continuous Normalizing Flows (CNFs) (Chen et al., 2018; Grathwohl et al., 2018) learn an unknown data distribution $p_T = p_{\mathrm{data}}(\mathbf{x})$ by transforming a simple base distribution $p_0$ into the data distribution via Neural ODEs from time 0 to $T$,

$$\text{forward}: \quad d\mathbf{x}_t = \mathbf{f}_\theta(\mathbf{x}_t, t)dt, \quad \mathbf{x}_0 \sim p_0 \tag{2}$$

$$\text{reverse}: \quad d\mathbf{x}_t = -\underbrace{\mathbf{f}_\theta(\mathbf{x}_t, t)}_{\text{velocity field}} dt, \quad \mathbf{x}_1 \sim p_1 \tag{3}$$

where $\mathbf{f}_\theta$ is a neural network that parametrizes the vector field driving the ODE. The training typically requires full simulation until time $T = 1$ and estimating log-likelihoods via the instantaneous change-of-variables formula. Recently, Flow Matching (FM) (Lipman et al., 2023; Tong et al., 2023) has emerged as an efficient method for training CNFs. The key idea is to construct a smooth probability path $(p_t)_{t \in [0,1]}$, generated by an underlying vector field $u_t(\mathbf{x}_t)$ that interpolates between $p_0$ and $p_T$,

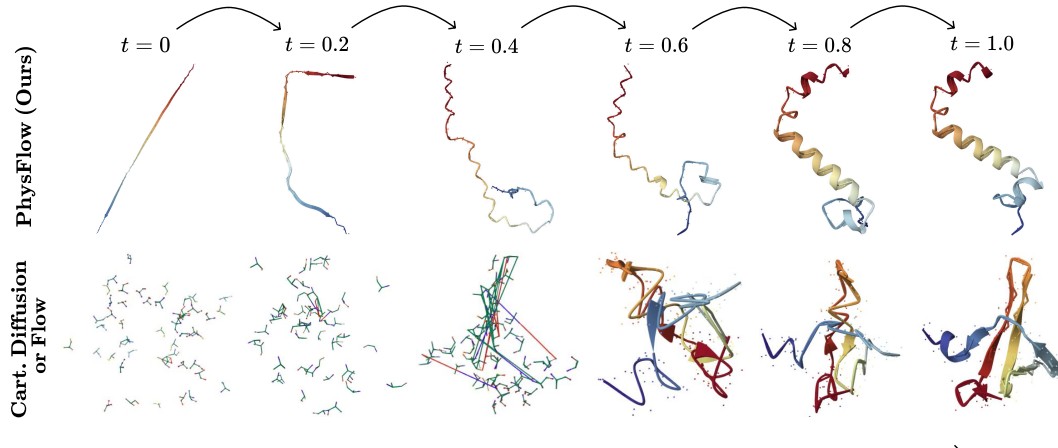

Figure 1: **Generation by PhysFlow and cartesian diffusion/flow-based methods**. Inference trajectories for unconditional monomer generation are compared between PhysFlow and cartesian diffusion/flow-based methods.

yielding a simulation-free training objective. However, this vanilla FM objective is intractable in practice, since the closed-form velocity field $u_t(\mathbf{x}_t)$ that generates $p_t$ is generally unavailable. To address this, we instead use conditional vector fields $u_t(\mathbf{x}_t|\mathbf{z})$, which define a conditional probability path $p_t(\mathbf{x}_t|\mathbf{z})$. The unconditional probability path can then be recovered by marginalizing over the conditioning variables $\mathbf{z}$. The resulting objective becomes tractable and the model can be trained by regressing on the conditional vector field with

$$\mathcal{L}_{\text{CFM}} = \mathbb{E}_{t,\mathbf{x}_t,\mathbf{z}}\Big[||\mathbf{f}_\theta(\mathbf{x}_t, t) - u_t(\mathbf{x}_t|\mathbf{z})||^2\Big], \qquad t \in \mathcal{U}[0,1], \quad \mathbf{z} \sim q(\mathbf{z}). \tag{4}$$

For an insightful review on flow-matching, we refer to Lipman et al. (2024).

**Diffusion and Flow models on proteins**  The earlier applications of diffusion and flow models to protein distributions operated directly over the Cartesian coordinates of the backbone (Watson et al., 2023; Lin and AlQuraishi, 2023; Ingraham et al., 2023; Geffner et al., 2025), while later models applied them in the space of translation-rotation frames (Yim et al., 2023a; Huguet et al., 2024; Campbell et al., 2024; Yim et al., 2023b; Bose et al., 2024) thereby enforcing SE(3) invariance. In all of these models, the forward process destroys the backbone into detached floating residues that can have intersecting linear trajectories with other residues, leading to clashes. In FoldingDiffusion (Wu et al., 2024), the noising is applied to backbone angles, which leads to a randomly oriented but intact backbone; however, the residues can still collide. In contrast to these methods, we leverage a physics-inspired noising process that unfolds proteins into secondary chains, whilst preserving bonds and avoiding clashes (as we describe next). See Table 1 for a comparative overview.

## 3   HOW REALISTIC IS THE GENERATIVE PROCESS?

**Linear diffusion and flow models are blind to protein physics.**  In linear diffusion and flow models, the forward noising process is typically defined as a linear Ornstein–Uhlenbeck process (Yim et al., 2023b; Bose et al., 2024; Santos and Lin, 2023) in $\mathbb{R}^n$, yielding closed-form transition kernels for the forward marginal $p_t$. While computationally convenient, this process overlooks the physical realism of proteins (Outeiral et al., 2022; Chakravarty et al., 2023; Uversky, 2002), treating residues as independent points rather than physical entities. Consequently, it *destroys* structural integrity by decorrelating residues and inducing silent collisions, thereby imposing an undesirable inductive bias, as illustrated in Fig. 5 for Cartesian diffusion and flow-based models. This inductive bias in the forward process (Vastola, 2025) is mirrored in the reverse generative process (Ho et al., 2020; Song et al., 2021b;a), making the model prone to generating structures with steric clashes, incurring violations of fundamental bond length and angle constraints (Anand et al., 2025) as well as biases (Lu et al., 2025) that hinder the recovery of consistent secondary and tertiary motifs.

**Physics-driven non-linear protein noising.**    To overcome these issues while maintaining physically meaningful correlations (Brini et al., 2020) during the noising process, we argue that a physically plausible generative (reverse) process requires a carefully designed forward process that *avoids collisions and preserves topological integrity*. Such a forward process is inherently non-linear, since structural constraints, angular dependencies, and other effects cannot be captured by linear dynamics (Frauenfelder, 2010).

In the following section 4, we show how to construct such a physics-driven non-linear noising process by drawing inspiration from *classical physics*, and how this process can be seamlessly combined with the flow-matching paradigm on SE(3) to model the invariant distribution of protein backbones.

## 4    METHOD

Our goal is to learn an SE(3)-invariant density $\rho_t$ using flows. This requires obtaining a pushforward from an initial SE(3)-invariant distribution $\rho_0$ to the empirical protein distribution $\rho_1$. Translation invariance can be enforced by centering each protein at its center of mass (Rudolph et al., 2021), which yields an invariant measure on $\mathrm{SE}(3)_0^N$ (subgroup of $\mathrm{SE}(3)^N$), for a protein of length $N$. Since $\mathrm{SE}(3)_0^N$ forms a product group, it admits a decomposition that enables flows to be constructed residue-wise. Consequently, a flow on $\mathrm{SE}(3)_0^N$ can be realized by combining independent flows over the residues. We provide a short recap about SE(3) lie groups in Appendix A.

**Decomposing** SE(3) **into** SO(3) **and** $\mathbb{R}^3$    SE(3) is a Lie group that can be represented as a semidirect product, $\mathrm{SE}(3) \cong \mathrm{SO}(3) \ltimes (\mathbb{R}^3, +)$. Accordingly, the metric on SE(3) can be decomposed as $\langle \mathfrak{r}, \mathfrak{r}' \rangle_{\mathrm{SE}(3)} = \langle \mathfrak{r}, \mathfrak{r}' \rangle_{\mathrm{SO}(3)} + \langle \mathbf{x}, \mathbf{x}' \rangle_{\mathbb{R}^3}$ (Bose et al., 2024). This choice allows the SE(3)-invariant measure to be decomposed into an independent SO(3)-invariant measure and a measure proportional to the Lebesgue measure on $\mathbb{R}^3$ (Pollard, 2002). Consequently, we can construct independent flows on SO(3) and $\mathbb{R}^3$. In the following, we describe how to design unfolding flows within this framework.

### 4.1    BUILDING SO(3) FLOWS

We aim to construct a flow on SO(3) that transports a prior distribution $\rho_0$ (e.g. $\mathcal{U}(\mathrm{SO}(3))$) to a target data distribution $\rho_1$. Following the parametrization strategy outlined in Bose et al. (2024), we define a time-dependent vector field $v_t(\mathbf{r}_t | \mathbf{r}_0, \mathbf{r}_1)$ that lies in the tangent space $\mathcal{T}_{\mathbf{r}_t} \mathrm{SO}(3)$ and depends on both the initial $\mathbf{r}_0$ and the target $\mathbf{r}_1$ (Tong et al., 2023). We adopt an approach based on geodesics on SO(3), by defining the interpolation between $\mathbf{r}_0$ and $\mathbf{r}_1$ along the geodesic as $\mathbf{r}_t = \exp_{\mathbf{r}_0}(t \log_{\mathbf{r}_0}(\mathbf{r}_1))$.

However, due to the computational cost and numerical instability associated with evaluating the exponential and logarithm maps on SO(3), controlling the approximation error can be expensive (Al-Mohy and Higham, 2012). To mitigate this, we adopt a numerical strategy proposed by Bose et al. (2024), which involves converting $\mathbf{r}_1$ into its axis-angle representation and applying parallel transport to efficiently compute $\log_{\mathbf{r}_0}(\mathbf{r}_1)$.

Given $\mathbf{r}_t$, we can utilize the Riemannian flow matching framework (Chen and Lipman, 2023) to build a conditional flow, giving $v_t = \dot{\mathbf{r}}_t$, which entails estimating the gradient of $\mathbf{r}_t$ at time $t$. To mitigate this issue, we instead compute the element of $\mathfrak{so}(3)$ corresponding to the relative rotation between $\mathbf{r}_0$ and $\mathbf{r}_t$, given as $\mathbf{r}_t^\top \mathbf{r}_0$ (Bose et al., 2024). Taking the matrix logarithm of this relative rotation and dividing by $t$ yields a skew-symmetric matrix in $\mathfrak{so}(3)$, representing the velocity vector pointing toward the target $\mathbf{r}_1$. We can then multiply by $\mathbf{r}_t$ to parallel transport over the tangent space. Finally, expressing these operations as $\log_{\mathbf{r}_t}(\mathbf{r}_0)$, we obtain the final velocity field as

$$v_t = \dot{\mathbf{r}}_t = \frac{\log_{\mathbf{r}_t}(\mathbf{r}_0)}{t} \tag{5}$$

### 4.2    UNFOLDING IN $\mathbb{R}^3$

We seek to define a translation-invariant unfolding process on $\mathbb{R}^3$ by leveraging the angular representation. Unlike raw Cartesian coordinates, the angular representation is inherently translation-invariant, making it a natural choice for this purpose. We formalize this in the following proposition:

**Proposition 1** (Translation Invariance of Angular Representation). *Let* $\mathbf{x} \in \mathbb{R}^{3N}$ *denote the backbone Cartesian coordinates of a protein with $N$ residues, and let $\mathbf{z}(\mathbf{x})$ be its angular representation. Let $\mathcal{T}_\mathbf{t} : \mathbb{R}^{3N} \to \mathbb{R}^{3N}$ denote a translation by vector $\mathbf{t} \in \mathbb{R}^3$ applied to all residues. Then, the angular representation is translation-invariant:*

$$\mathbf{z}(\mathcal{T}_\mathbf{t} \circ \mathbf{x}) = \mathbf{z}(\mathbf{x}), \quad \forall \, \mathbf{t} \in \mathbb{R}^3. \tag{6}$$

We provide the proof in Appendix C. Moreover, since there exists a bijection between the angular and Cartesian domains (Parsons et al., 2005; Derevyanko and Lamoureux, 2018; Ingraham et al., 2019), any flow defined in angular space induces a corresponding flow in Cartesian space that automatically preserves translation invariance.

We assume a second-order Decay-Hamiltonian (Desai et al., 2021; Neary and Topcu, 2023) unfolding process over the angular space $\mathbf{z}$,

$$\text{unfolding:} \quad \begin{cases} \dot{\mathbf{z}}_t & = \mathbf{v}_t \\ \dot{\mathbf{v}}_t & = f_{\text{unfold}}(\mathbf{z}_t, \mathbf{v}_t) = \underbrace{-\nabla_{\mathbf{z}_t} U(\mathbf{z}_t)}_{\text{potential force}} - \underbrace{K(\mathbf{v}_t)}_{\text{drag force}}, \end{cases} \tag{7}$$

which follows an Ornstein-Uhlenbeck energy potential $U(\mathbf{z}_t)$ and a $K(\mathbf{v}_t)$ friction. Our main goal is to unfold into a secondary structure without unrealistic residue collisions. We propose an augmented potential

$$U(\mathbf{z}_t) = k_1 U_{\text{target}}(\mathbf{z}_t) + k_2 U_{\text{repulsion}}\big( \text{nerf}(\mathbf{z}_t)\big) \tag{8}$$

that incorporates both Ornstein-Uhlenbeck attraction to $\mathbf{z}_{\text{target}}$ (e.g. can be a secondary structure chain such as $\beta$-sheet, $\alpha$-helix, etc.) and Coulomb-like repulsion

$$U_{\text{target}}(\mathbf{z}_t) = \frac{1}{2} \sum_i \big( \mathbf{z}_{i,t} - \mathbf{z}_{i,\text{target}} \big)^2, \quad U_{\text{repulsion}}(\mathbf{x}_t) = \frac{1}{2} \sum_{i \neq j} \frac{1}{||\mathbf{x}_{i,t} - \mathbf{x}_{j,t}||}, \tag{9}$$

where $i, j$ denote the residue indices along the backbone, $\mathbf{z}_{\text{target}} \sim p_0(\mathbf{z})$, $k_1, k_2$ are the respective weights for the potentials, and the Coulomb repulsion term is based on Euclidean distance. The target energy is a quadratic potential which forces the protein towards the target state, while the repulsion represents a Coulomb barrier that prevents collisions. We use JAX autodiff to compute the gradient $\nabla_{\mathbf{z}_t} U(\mathbf{z}_t)$, which also flows through the cartesian reconstruction nerf[1] (Bradbury et al., 2018).

**Friction.** The potential energy $U(\mathbf{z}_t)$ creates highly oscillatory dynamics requiring stabilization. We introduce a drag force $K(\mathbf{v}_t) = -\gamma \mathbf{v}_t$ where $\gamma > 0$ is a drag coefficient representing environmental resistance. This dampening term counteracts strong accelerations and stabilizes the system, ensuring convergence to $\mathbf{z}_{\text{target}}$ as $t \to \infty$.

**Terminal distributions.** We assume the final target state to be a secondary structure linear chain predefined as linear $\beta$-sheet. Denoting $\mathbf{u} = [\mathbf{z}, \mathbf{v}]$, the terminal distribution is defined as:

$$p_0(\mathbf{u}) = \underbrace{\mathcal{N}(\mathbf{z}|\boldsymbol{\mu}_\beta, \sigma_\beta^2 I)}_{\text{position } p_0(\mathbf{z})} \cdot \underbrace{\mathcal{N}(\mathbf{v}|\mathbf{0}, \sigma_v^2 I)}_{\text{velocity}}, \tag{10}$$

where there is small variation $\sigma_\beta^2$ around the $\beta$-sheet angles $\mu_\beta$ and the initial velocity is Gaussian. We also assume a data distribution

$$p_1(\mathbf{u}) = p_{\text{data}}(\mathbf{z}) \cdot \mathcal{N}(\mathbf{v}|\mathbf{0}, \sigma_v^2 I), \tag{11}$$

from which we have observations without velocities, and we augment the data with Gaussian velocities of variance $\sigma_v^2$.

---

[1] github.com/PeptoneLtd/nerfax

Figure 2: **PhysFlow Model Pipeline** ($f_\eta$). The model takes as input a noised structural state together with the sequence information. These are first processed independently by a structure encoder and a sequence encoder. The resulting representations are then integrated through a combiner module, after which a structure decoder predicts both the velocity field and the auxiliary predictions.

**Computing Forward Simulations.** Due to the presence of the Coulomb repulsion potential, computing the forward transition kernel in closed form is analytically intractable. As a result, we rely on forward simulations. Specifically, for any given time step $t$ with a given data point $\mathbf{u}_1 \sim p_1(\mathbf{u})$ and the prior $\mathbf{u}_0 \sim p_0(\mathbf{u})$, the forward transition distribution is given by:

$$p(\mathbf{u}_t|\mathbf{u}_1, \mathbf{u}_0) = \mathcal{N}\left(\begin{bmatrix} \mathbf{z}_t \\ \mathbf{v}_t \end{bmatrix}; \begin{bmatrix} \mu_{\mathbf{z}_t} \\ \mu_{\mathbf{v}_t} \end{bmatrix}, \begin{bmatrix} \sigma_z^2 \\ \sigma_v^2 \end{bmatrix} \mathrm{I}\right), \quad \begin{bmatrix} \mu_{\mathbf{z}_t} \\ \mu_{\mathbf{v}_t} \end{bmatrix} = \begin{bmatrix} \mathbf{z}_1 \\ \mathbf{v}_1 \end{bmatrix} - \int_t^1 \begin{bmatrix} \dot{\mathbf{z}}_t \\ \dot{\mathbf{v}}_t \end{bmatrix} d\tau \qquad (12)$$

where $\sigma_{z,v}$ is the variance typically assumed to be fixed across intermediate distributions and chosen based on the data, and $\mu_{\mathbf{z}_t, \mathbf{v}_t}$ is obtained by solving a second-order ODE system that governs the dynamics under the potential. Recall that due to bijection, we can obtain the flow in cartesian space by applying $\mathrm{nerf}$ as, $p_{\mathrm{cart}}(\mathbf{u}_t|\mathbf{u}_1, \mathbf{u}_0) = p(\mathrm{nerf}(\mathbf{u}_t)|\mathrm{nerf}(\mathbf{u}_1), \mathrm{nerf}(\mathbf{u}_0))$, where $\mathbf{u}_1 \sim p_1(\mathbf{u})$, and $\mathbf{u}_0 \sim p_0(\mathbf{u})$ (see Appendix C.2 for details). Given a dataset, we can simulate forward trajectories and store the resulting sample trajectories, which can then be used to train a model to learn the reverse process via conditional flow matching (Lipman et al., 2023) in $\mathbb{R}^3$.

## 4.3 MODEL ARCHITECTURE

We employ a multimodal network $f_\eta(\mathbf{x}_t, \mathbf{r}_t, \bar{a}, t, s)$ similar to (Huguet et al., 2024), that takes as input the noised translation–rotation frames, sequence information, and the time step, and predicts the corresponding velocity fields along with auxiliary predictions. The architecture comprises the following modules:

**Structure Encoder.** To encode structural information, we adopt the invariant point attention (IPA) transformer (Jumper et al., 2021), an SE(3)-equivariant architecture. The IPA is highly flexible: it can both consume and generate structures as $N$ rigid frames, while simultaneously producing single and pairwise representations of the input.

**Sequence Encoder.** We incorporate sequence information using the 650M-parameter variant of the pre-trained ESM2 sequence model (Lin et al., 2022), which equips our framework with strong sequence modeling capabilities. The architecture outputs both single and pair representations for a given amino acid sequence, conceptually analogous to the representations produced by the structure encoder. To enable both conditional and unconditional training, sequence information $a$ is randomly masked with probability $p_{\mathrm{uncond}}$,

$$\bar{a} = \mathbb{1}(p_{\mathrm{uncond}} < 0.5) \cdot a + \left(1 - \mathbb{1}(p_{\mathrm{uncond}} < 0.5)\right) \cdot \emptyset \qquad (13)$$

where $p_{\mathrm{uncond}} \sim \mathcal{U}[0,1]$, and $\mathbb{1}(p_{\mathrm{uncond}} < 0.5)$ denotes the indicator function that ensures sequence information is available $50\%$ of the time.

**Combiner Module and Structure Decoder.** After encoding both the input structure and sequence, we construct a joint representation of the single and pair embeddings, following Huguet et al. (2024), which iteratively updates them via triangular self-attention. These updated representations are then fed into the decoder, parameterized as an IPA transformer, to predict the velocity fields along with auxiliary outputs.

The full model pipeline is illustrated in Figure 2, and the training objective is described next.

## 5 LEARNING OBJECTIVE

**Flow matching Loss.** We utilize the conditional flow matching (CFM) loss (Lipman et al., 2023; Tong et al., 2023) to learn the velocity fields. Specifically, for SO(3) and $\mathbb{R}^3$, the corresponding CFM objectives are defined as

$$\mathcal{L}_{\text{SO(3)}} = \mathbb{E}_{t,\mathbf{r}_t,\mathbf{r}_0,\mathbf{r}_1} \left\| \frac{\log_{\mathbf{r}_t}(\mathbf{r}_0)}{t} - \hat{v}_{\mathbf{r}_t}(t) \right\|_{\text{SO(3)}}^2, \text{ where } \mathbf{r}_t \sim p(\mathbf{r}_t|\mathbf{r}_1,\mathbf{r}_0) \quad (14)$$

$$\mathcal{L}_{\mathbb{R}^3} = \mathbb{E}_{t,\mathbf{x}_t,\mathbf{x}_0,\mathbf{x}_1} \left\| v(\mathbf{x}_t|\mathbf{x}_1,\mathbf{x}_0) - \hat{v}_{\mathbf{x}_t}(t) \right\|_2^2, \text{ where } \mathbf{x}_t \sim p(\mathbf{x}_t|\mathbf{x}_1,\mathbf{x}_0) \quad (15)$$

where $\hat{v}_{\mathbf{x}_t}(t), \hat{v}_{\mathbf{r}_t}(t)$ denotes the velocity fields predicted by the model, $t \sim \mathcal{U}[0,1]$, $(\mathbf{x}_0,\mathbf{x}_1) \sim q(\mathbf{x}_0,\mathbf{x}_1) = p(\text{nerf}(\mathbf{z}_0))p(\text{nerf}(\mathbf{z}_1))$, $(\mathbf{r}_0,\mathbf{r}_1) \sim q(\mathbf{r}_0,\mathbf{r}_1) = \rho(\mathbf{r}_0)\rho(\mathbf{r}_1)$ follow independent coupling, with $\log_{\mathbf{r}_t}(\mathbf{r}_0)/t$ denoting the velocity field on SO(3), and $v(\mathbf{x}_t|\mathbf{x}_1,\mathbf{x}_0)$ the corresponding velocity field on $\mathbb{R}^3$ obtained via the unfolding process.

**Look-Ahead (LA) Loss.** To help the model learn non-linear unfolding trajectories, we introduce a novel look-ahead loss that penalizes the model for inaccurate predictions of future states. This loss is conceptually similar to the one proposed in Zhou et al. (2025). Specifically, it is defined as

$$\mathcal{L}_{\text{LA}} = \mathbb{E}_{t,s} \left[ \frac{1}{4N} \sum_{n=1}^{N} \sum_{a \in \Omega} ||a_n^{(s)} - \hat{a}_n^{(s)}||^2 \right], \text{ where } t \sim \mathcal{U}[0,1], \text{ and } s \sim \mathcal{U}[t,1] \quad (16)$$

where $a_n^{(s)} = \zeta_{\text{frame-to-atom}}(\mathbf{r}_s, \mathbf{x}_s)$, $\hat{a}_n^{(s)} = \zeta_{\text{frame-to-atom}}(\hat{\mathbf{r}}_s, \hat{\mathbf{x}}_s)$, $\zeta_{\text{frame-to-atom}}$ representing transformation from frames to coordinates, and $\Omega = \{N, C, O, C_\alpha\}$.

**Auxiliary Losses.** To encourage the model to capture fine-grained local characteristics of protein structures, we incorporate two auxiliary losses at the final step of generation (Yim et al., 2023b). The first is a direct mean-squared error (MSE) loss on predicted terminal backbone atom positions, denoted by $\mathcal{L}_{\text{bb}}$. The second, $\mathcal{L}_{\text{2D}}$ is analogous to the distogram loss used in AF2,

$$\mathcal{L}_{\text{bb}} = \frac{1}{4N} \sum_{n=1}^{N} \sum_{a \in \Omega} ||a_n^{(1)} - \hat{a}_n^{(1)}||^2, \quad \mathcal{L}_{\text{2D}} = \frac{1}{Z} \sum_{n,m=1}^{N} \sum_{a,b \in \Omega} \mathbb{1}(d_{ab}^{nm} < 0.6)||d_{ab}^{nm} - \hat{d}_{ab}^{nm}||^2 \quad (17)$$

where $a_n^{(1)} = \zeta_{\text{frame-to-atom}}(\mathbf{r}_1, \mathbf{x}_1)$, $\hat{a}_n^{(1)} = \zeta_{\text{frame-to-atom}}(\hat{\mathbf{r}}_1, \hat{\mathbf{x}}_1)$, $\Omega = \{N, C, O, C_\alpha\}$, $d_{ab}^{nm} = ||a_n^{(1)} - b_m^{(1)}||$, $\hat{d}_{ab}^{nm} = ||\hat{a}_n^{(1)} - \hat{b}_m^{(1)}||$, and $Z = (\sum_{n,m=1}^{N} \sum_{a \in \Omega} \mathbb{1}(d_{ab}^{nm} < 0.6)) - N$. Here $\mathbb{1}(d_{ab}^{nm} < 0.6)$ is an indicator function that restricts the loss to atom pairs within 0.6nm.

We combine the flow-matching loss with the auxiliary and look-ahead losses to get the final loss:

$$\mathcal{L} = \mathcal{L}_{\text{SO(3)}} + \mathcal{L}_{\mathbb{R}^3} + \mathcal{L}_{\text{LA}} + \lambda \cdot \mathbb{1}(t > 0.75)(\mathcal{L}_{\text{bb}} + \mathcal{L}_{\text{2D}}) \quad (18)$$

where the auxiliary losses are applied only during the later stages of generation ($t > 0.75$), as indicated by the indicator function $\mathbb{1}(t > 0.75)$, with a scaling $\lambda$.

## 6 EXPERIMENTS

**Tasks.** We evaluate PhysFlow's generative capabilities in two settings: (i) unconditional monomer backbone structure generation across proteins of varying lengths (Section 6.1), and (ii) sequence conditioned monomer folding, where the task is to generate the corresponding protein structure given an input sequence (Section 6.2).

**Data.** We train PhysFlow on monomers[2] with lengths between 60 and 512 residues and structural resolution $< 5\text{Å}$, obtained from the Protein Data Bank (PDB) (Berman et al., 2000). Following the filtering criteria outlined in Yim et al. (2023b); Bose et al. (2024), we obtain a dataset of 24,003 proteins. For each protein, we run forward unfolding simulations and store the resulting trajectories for model training. More details can be found in Appendix B.

---

[2]The oligomeric state is determined from the metadata provided in the mmCIF file.

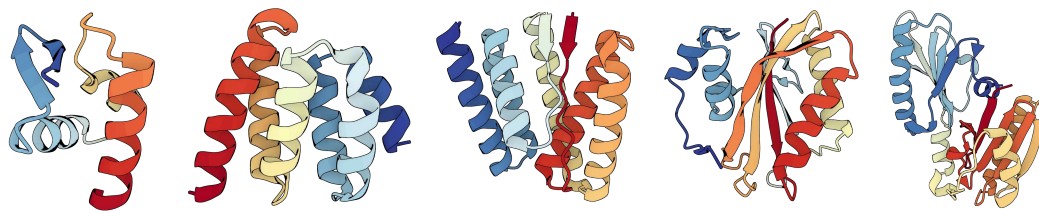

Figure 3: **PhysFlow Samples**. Designable backbones generated unconditionally by PhysFlow model.

Table 2: **PhysFlow Unconditional Backbone Generation Performance**. Comparison of designability, novelty, secondary structure composition, and diversity of generated protein structures. RFDiffusion uses pretraining, while FoldFlow-2, ESM-3, Chroma, and Proteína leverage additional/different datasets for training or fine-tuning. Best results are in **bold**, second-best are underlined.

| Method | Designability | Novelty | | Diversity | | Sec. Struct % |
|---|---|---|---|---|---|---|
| | Frac. $< 2\text{Å}$ ($\uparrow$) | TM-Frac. $< 0.3$ ($\uparrow$) | avg. max TM ($\downarrow$) | Cluster ($\uparrow$) | TM-Sc. ($\downarrow$) | $(\alpha/\beta)$ |
| RFDiffusion[†] | 0.94 | $0.116 \pm 0.020$ | $0.449 \pm 0.012$ | 0.17 | 0.25 | 64.3/17.2 |
| FoldFlow-2[†] | 0.97 | $0.368 \pm 0.031$ | $0.363 \pm 0.012$ | 0.34 | 0.20 | 82.7/2.0 |
| Proteína[†] | 0.99 | $0.384 \pm 0.011$ | $0.313 \pm 0.001$ | 0.30 | 0.39 | 62.2/9.9 |
| ESM3[†] | 0.22 | $0.234 \pm 0.031$ | $0.421 \pm 0.018$ | 0.48 | 0.42 | 64.5/8.5 |
| Chroma[†] | 0.57 | $0.214 \pm 0.033$ | $0.412 \pm 0.011$ | 0.13 | 0.27 | 69.0/12.5 |
| Genie | 0.55 | $0.120 \pm 0.021$ | $\mathbf{0.434} \pm 0.016$ | 0.27 | **0.22** | 72.7/4.8 |
| FrameDiff | 0.40 | $0.020 \pm 0.009$ | $0.542 \pm 0.046$ | 0.31 | 0.23 | 64.9/11.2 |
| FrameFlow | 0.60 | $0.110 \pm 0.011$ | $0.481 \pm 0.026$ | **0.36** | **0.22** | 55.7/18.4 |
| FoldFlow (CFM) | 0.65 | $\underline{0.188} \pm 0.025$ | $0.460 \pm 0.020$ | 0.22 | 0.24 | 86.1/1.2 |
| FoldFlow (OT) | 0.77 | $0.181 \pm 0.015$ | $0.452 \pm 0.024$ | 0.23 | 0.26 | 84.1/3.2 |
| PhysFlow | **0.81** | $\mathbf{0.197} \pm 0.021$ | $\underline{0.450} \pm 0.029$ | 0.34 | **0.22** | 82.2/5.7 |

**Competing methods.** We compare PhysFlow against a diverse set of protein structure generation methods. These include diffusion-based approaches such as FrameDiff (Yim et al., 2023b), RFDiffusion (Watson et al., 2023), Chroma (Ingraham et al., 2023), and Genie (Lin and AlQuraishi, 2023); flow-based approaches such as FoldFlow's (Bose et al., 2024; Huguet et al., 2024), FrameFlow (Yim et al., 2023a), and Proteína (Geffner et al., 2025); as well as ESM3 (Hayes et al., 2025), a state-of-the-art masked language model that is also capable of producing protein structures.

6.1 UNCONDITIONAL MONOMER PROTEIN GENERATION

We evaluate PhysFlow's unconditional generation capabilities using several benchmarking metrics: **Designability** (the fraction of generated proteins that are designable), **Novelty**, **Diversity**, and **secondary structure composition**. Detailed definitions of these metrics are provided in App. B.2. To assess performance across sequence lengths, we generate 50 samples for each target length in $\{100, 150, 200, 250, 300\}$ and compute the metrics on the resulting sets.

**Improved Unconditional Generation Results.** Table 2 summarizes the performance of PhysFlow across multiple metrics against existing baselines, with representative uncurated samples shown in Figure 3. PhysFlow generates the most designable samples, producing structures that can be refolded by ESMFold to within $< 2\text{Å}$. It also achieving greater novelty than FrameDiff, FrameFlow, and FoldFlow, despite relying on the same training dataset and without additional datasets, pre-training, or fine-tuning. By contrast, models such as RFDiffusion, FoldFlow-2, Proteina, and ESM3 exploit different datasets alongside pre-training or fine-tuning, making direct comparisons unfair.

6.2 SEQUENCE CONDITIONED MONOMER FOLDING

We benchmark PhysFlow on sequence-conditioned monomer folding, where the task is to predict a folded protein structure from its amino acid sequence. For computational feasibility, our evaluation focuses exclusively on monomers. Training and test sets are curated from 24,003 filtered monomers in the PDB, stratified by cluster and sequence length $\ell_{prot}$ (see App. B.3 for details). We

Table 3: **Ablation Studies and Monomer Folding Performance**. (*Left*) Ablation study showing the effect of varying the number of time steps during inference on designability and novelty metrics. (*Right*) Evaluation on the test set for sequence-conditioned monomer folding. [†]ESMFold is trained on ∼65 million UniRef sequences, while PhysFlow achieves the lowest RMSD.

| Time-steps | Frac. $< 2\text{Å}$ ($\uparrow$) | TM-Frac. $< 0.3$ ($\uparrow$) |
|---|---|---|
| 50 | 0.805 | $0.201 \pm 0.024$ |
| 100 | 0.817 | $0.197 \pm 0.021$ |
| 200 | 0.811 | $0.199 \pm 0.011$ |

| Method | RMSD (Å) | |
|---|---|---|
| | $\ell_{prot} < 200$ | $\ell_{prot} \geq 200$ |
| ESMFold[†] | $1.64 \pm 2.73$ | $2.82 \pm 3.71$ |
| MultiFlow | $17.41 \pm 2.17$ | $20.63 \pm 2.75$ |
| FoldFlow-2 | $13.52 \pm 1.84$ | $17.88 \pm 2.13$ |
| PhysFlow | $\mathbf{11.08} \pm 1.02$ | $\mathbf{15.11} \pm 1.79$ |

compare PhysFlow against the gold-standard ESMFold (Lin et al., 2023), as well as state-of-the-art sequence-augmented structure generation methods FoldFlow-2 and MultiFlow (Campbell et al., 2024), both retrained solely on our training set without extra pre-training or fine-tuning.

**Superior Results in Monomer Folding.** Table 3 compares PhysFlow against baseline methods. It is worth noting that ESMFold was trained on a substantially larger dataset of ∼65 million sequence–structure pairs from UniRef (Suzek et al., 2015), making direct comparisons with our setting less equitable. Nevertheless, PhysFlow achieves the lowest RMSD among baselines such as FoldFlow-2 and MultiFlow which have been trained on same dataset, for both short and long protein sequences, thereby demonstrating the effectiveness of our approach.

## 7 ABLATION STUDIES

**Runtime Complexity.** We evaluate the runtime complexity of our forward simulation (see Section 4.2) as a function of protein length. Figure 4 illustrates the scaling behavior in terms of CPU time (seconds). As expected, longer proteins require more time for forward simulation. However, the overall runtime remains modest, and the process can be further accelerated through parallelization and multi-processing techniques, not posing a computational bottleneck in our framework.

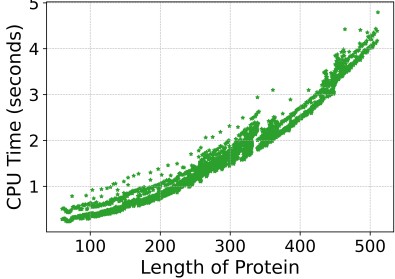

Figure 4: Runtime Complexity.

**Effect of varying time steps.** We evaluate the impact of the number of time steps used in solving the reverse process on the designability and novelty. Table 3 show that PhysFlow achieves strong performance even with relatively few steps, while maintaining stability as the number of steps increases.

**Effect of varying $k_1, k_2$ energy weighting.** We conducted an ablation study to assess the effect of varying the energy weightings $k_1, k_2$ that govern the forward dynamics of our model. As shown in Figure 5, different weighting schemes yield distinct forward processes, characterized by changes in the number of steric clashes across pairwise residue distances. Within the PhysFlow framework, we find that increasing the weight on Coulombic repulsion ($k_2$) reduces the number of clashes.

## 8 CONCLUSION AND FUTURE WORK

We introduce PhysFlow, a generative model that incorporates a physics-informed noising process based on Hamiltonian dynamics to model protein backbones. Unlike the linear forward processes, our noising preserves structural integrity and prevents clashes, providing a more physically realistic inductive bias. Combined with flow matching on $SE(3)$, this enables accurate backbone generation while respecting the geometry of protein structures. PhysFlow achieves state-of-the-art performance in unconditional generation of novel and designable proteins, and in folding monomer sequences. Extending our model beyond monomers and using additional datasets such as AFDB (Varadi et al., 2022) presents a compelling avenue for further research.

## Reproducibility Statement

Our work utilizes fully open-source data from the Protein Data Bank, and we follow the preprocessing steps described in the baseline methods, which are also publicly available in their respective papers. Details of the model architecture, hyperparameters, and training setup are provided in the appendix. In addition, code snippets illustrating the forward simulation procedure are included in the appendix for clarity and reproducibility. We will make our code public upon acceptance.

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

## A  SHORT REVIEW OF LIE AND SE(3) GROUPS

### A.1  LIE GROUPS

Symmetries refer to transformations of an object that preserve certain structural properties. When these symmetries form a continuous set and are equipped with a composition operation that satisfies the group axioms, they define a Lie group. Formally, a group is a set $G$ together with a binary operation $\circ : G \times G \to G$ that satisfies the following properties:

- Associativity: $(x \circ y) \circ z = x \circ (y \circ z), \forall\, x, y, z \in G$
- Identity: There exists an element $e \in G$, such that $x \circ e = e \circ x = x, \forall\, x \in G$
- Inverses: For every $x \in G$, there exists $x^{-1} \in G$, such that $x \circ x^{-1} = x^{-1} \circ x = e$

A Lie group extends this structure by also being a smooth manifold, meaning that the group operations—multiplication $(x, y) \to xy, \forall\, x, y \in G$ and inversion $x \to x^{-1}$ are smooth maps.

Given a $y \in G$, we define a diffeomorphism $L_y : G \to G, x \to yx$ known as left multiplication. Given a vector field $X$ on the group $G$, we say that $X$ is left-invariant if it remains unchanged under the pushforward induced by left multiplication. Formally, this means:

$$L_y^* X = X, \forall\, y \in G \tag{19}$$

where $L_y^*$ denotes the differential (pushforward) of the left multiplication map. This pushforward naturally identifies the tangent spaces via:

$$\mathcal{T}_x G \xrightarrow{L_y^*} \mathcal{T}_{yx} G \tag{20}$$

As a consequence, a vector field is entirely determined by its value at the identity element $e \in G$ i.e. $\mathcal{T}_e$. We can further equip a real vector space $V$ with a bilinear operation known as the Lie bracket $[\cdot, \cdot] : V \times V \to V$, which satisfies two key properties:

- Antisymmetry: $[x, y] = -[y, x], \ \forall\, x, y \in V$
- Jacobi identity: $[x, [y, z]] + [y, [z, x]] + [z, [x, y]] = 0, \ \forall\, x, y, z \in V$

A vector space $V$ endowed with such a bracket is called a Lie algebra. For a Lie group $G$, the tangent space at the identity element $\mathcal{T}_e G$ naturally forms a Lie algebra. This Lie algebra is often denoted $\mathfrak{G}$. There exists a smooth, invertible map known as the exponential map $\exp : \mathfrak{G} \to G$, which maps elements of the Lie algebra to elements of the Lie group. Its inverse, when defined, is called the logarithmic map: $\log : G \to \mathfrak{G}$. These maps allow us to move between the Lie algebra and the Lie group, and in the case of matrix Lie groups, they coincide with the standard matrix exponential and logarithm.

Finally, we note that the set of all $n \times n$ non-singular (invertible) matrices forms a Lie group. The group operation is matrix multiplication, and this set can be viewed as a smooth manifold. This group is known as the *General Linear Group*, denoted $GL(n)$. Any closed subgroup of $GL(n)$ is referred to as a matrix Lie group, which is among the most widely studied and practically important classes of Lie groups. For matrix Lie groups, the exponential and logarithmic maps align with the matrix exponential and matrix logarithm, respectively. For a comprehensive introduction to Lie groups and their algebraic and geometric structures, we refer the reader to Hall (2013).

### A.2  SE(3): SPECIAL EUCLIDEAN GROUP IN 3 DIMENSIONS

One of the most extensively studied closed subgroups of $GL(n)$ is the Special Orthogonal Group in three dimensions, denoted $SO(3)$ This group consists of all $3 \times 3$ real rotation matrices, which preserve both lengths and orientations in three-dimensional space. More generally it can be defined as,

$$\mathrm{SO(n)} = \{\mathrm{R} \in GL(n) | \mathrm{R}^\top \mathrm{R} = \mathrm{I}, \det(\mathrm{R}) = 1\} \tag{21}$$

The condition $\mathrm{R}^\top \mathrm{R} = \mathrm{I}$ ensures that the transformation is orthogonal (i.e., distance-preserving), and $\det(\mathrm{R}) = 1$ guarantees that it is a proper rotation, excluding reflections. Moreover, translations in

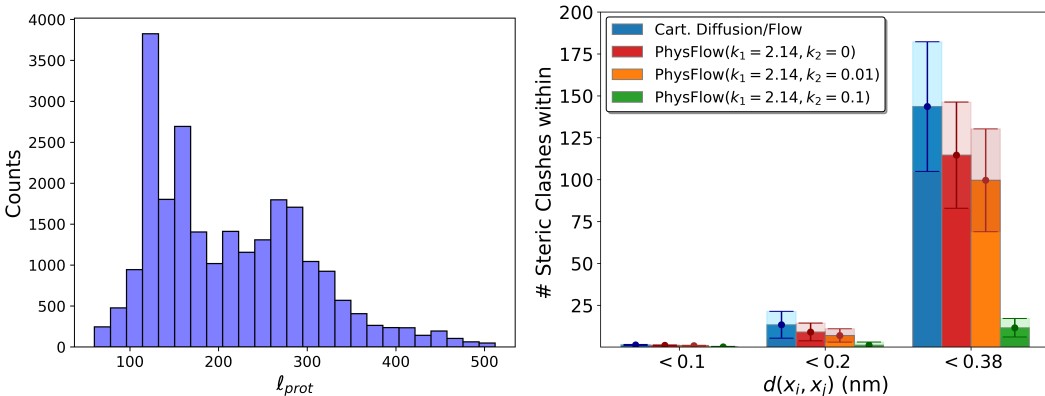

Figure 5: (*left*) Distribution of the sequence length of protein $\ell_{prot}$ in the obtained monomer dataset, (*right*) Average number of residue–residue steric collisions (defined as pairwise residue distance $< d(\mathbf{x}_i, \mathbf{x}_j)(\text{nm})$) observed across the forward trajectory of a protein of length 116, for Cartesian diffusion/flow baselines and variants of our PhysFlow method with different weightings $(k_1, k_2)$. We observe that the number of collisions decreases as the weight on Coulombic repulsion $(k_2)$ increases within the PhysFlow framework.

3D space can also be represented as a matrix Lie group. A translation by a vector $s \in \mathbb{R}^3$ can be expressed using a homogeneous transformation matrix of the form:

$$\begin{bmatrix} I & s \\ 0 & 1 \end{bmatrix} \tag{22}$$

The group operation corresponds to vector addition of the translation components: $s_1 + s_2$. This group of translations forms a matrix Lie group isomorphic to $(\mathbb{R}^3, +)$ the additive group of 3D vectors. Combining both the SO(3) and $\mathbb{R}^3$ groups, we can represent rigid transformations of objects in 3D space. This group is known as the Special Euclidean group in three dimensions, denoted SE(3), which can further decomposed as $\text{SE}(3) \cong \text{SO}(3) \ltimes (\mathbb{R}^3, +)$ under certain conditions. An element of SE(3) can be represented as a homogeneous transformation matrix:

$$\text{SE}(3) = \left\{ (r, s) = \begin{pmatrix} r & s \\ 0 & 1 \end{pmatrix} : r \in \text{SO}(3), s \in (\mathbb{R}^3, +) \right\} \tag{23}$$

Under a specific choice of Riemannian metric (Park and Brockett, 1994), the inner product on the Lie algebra $\mathfrak{se}(3)$ can be decomposed as $\langle \mathfrak{r}, \mathfrak{r}' \rangle_{\text{SE}(3)} = \langle r, r' \rangle_{\text{SO}(3)} + \langle s, s' \rangle_{\mathbb{R}^3}$, which forms the foundation of our model.

## B  ADDITIONAL EXPERIMENT DETAILS

### B.1  DATA

We train PhysFlow on monomers of length 60 to 512 and resolution better than 5Å obtained from the Protein Data Bank (PDB) (Berman et al., 2000). To ensure higher structural quality, we filtered the dataset to retain proteins with substantial secondary structure content. Similar to Yim et al. (2023b); Bose et al. (2024), we applied DSSP (Kabsch and Sander, 1983) and excluded monomers with more than 50% loop content, yielding a final dataset of 24,003 proteins belonging to 4532 clusters based on 40% sequence identity. The distribution of sequence lengths of protein $\ell_{prot}$ in this dataset is shown in Figure 5.

### B.2  BENCHMARKING METRICS

**Designability.** A protein backbone is considered designable if there exists an amino acid sequence. We evaluate designability following the protocol of Yim et al. (2023b). For each backbone generated

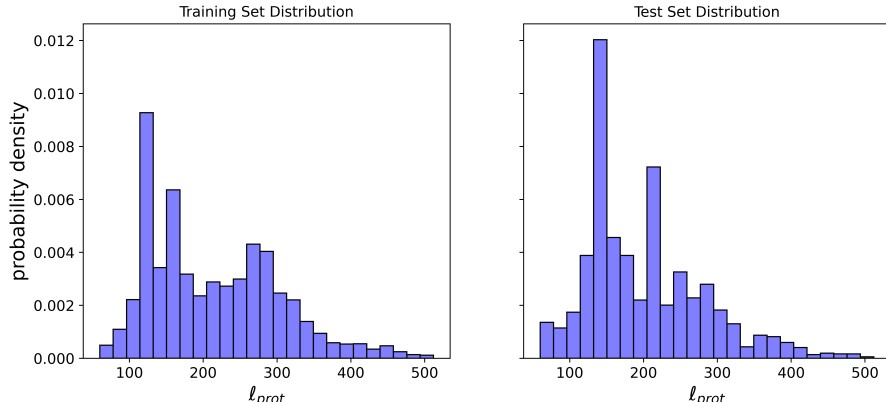

Figure 6: Training and Testing distribution for length of protein $\ell_{prot}$.

by a model, we sample sequences using ProteinMPNN (Dauparas et al., 2022) with a temperature of 0.1. Each sequence is then folded with ESMFold (Lin et al., 2023), and the resulting structure is compared against the original backbone using root mean square deviation (RMSD). A sample is classified as designable if its minimum RMSD—referred to as the self-consistency RMSD (scRMSD)—is below 2Å. The overall designability score of a model is defined as the fraction of generated backbones that satisfy this criteria.

**Diversity (TM-score & Cluster).**   We evaluate diversity using the measures described in Bose et al. (2024). We compute the average pairwise TM-score among designable samples for each protein length, and then aggregate these averages across all lengths. Since TM-scores range from 0 to 1, with higher values reflecting greater structural similarity, lower scores indicate higher diversity. The second metric is described in Huguet et al. (2024) which amounts to the number of generated clusters with a TM-score threshold of 0.5 (Herbert and Sternberg, 2008).

**Novelty (avg max TM).**   We quantify novelty using two metrics, following Bose et al. (2024). The first computes, for each designable generated protein, the maximum TM-score with respect to the training data. We then report the average of these maximum values across all designable samples. Lower scores indicate greater novelty.

**Novelty (TM-Frac).**   The second metric, introduced in Lin and AlQuraishi (2023), reports the fraction of generated proteins that are both designable and novel, i.e., whose maximum TM-score against the training set falls below a given threshold.

**Secondary Structure content.**   We analyze the secondary structure composition of designable backbones using Biotite's (Kunzmann and Hamacher, 2018) implementation of the P-SEA algorithm (Labesse et al., 1997). For each sample, we compute the proportions of $\alpha$-helices, $\beta$-sheets, and coils. These are reported as normalized fractions:

$$\frac{\alpha}{\alpha + \beta + c}, \quad \frac{\beta}{\alpha + \beta + c}, \quad \frac{c}{\alpha + \beta + c}$$

corresponding to helices, sheets, and coils, respectively.

### B.3 SEQUENCE CONDITIONED MONOMER FOLDING

We partition the dataset into a $0.9 : 0.1$ train–test split, selecting clusters randomly. Out of a total of 4,542 clusters, 4,087 are assigned to the training set and 455 to the test set. Figure 6 shows the distributions of protein sequence lengths in both sets, indicating that they are well-matched. Models are trained exclusively on the training set and evaluated on the test set to assess their performance in predicting folded structures.

### B.4 NEURAL NETWORK HYPERPARAMETERS

The training setup is summarized in Table 5, with network hyperparameters listed in Table 4. We adopt the "length batching" scheme from (Yim et al., 2023b), where each batch contains the same protein sampled at different time steps. The batch size varies approximately as $\text{num\_residues}^2/M$, with $M$ as a hyperparameter. PhysFlow is implemented in PyTorch and trained on a single H200 140GB NVIDIA GPU.

Table 4: Default hyperparameters for PhysFlow Model.

| Module | Hyperparameter | Meaning | Value |
|---|---|---|---|
| | Node embedding size | Node embedding dimension | 256 |
| | Edge embedding size | Edge embedding dimension | 128 |
| Structural Encoder | Hidden dimension | Hidden dimension | 256 |
| | Heads | Number of heads in the transformer | 8 |
| | Blocks | Number of transformer layers | 4 |
| | Node embedding size | Node embedding dimension | 256 |
| | Edge embedding size | Edge embedding dimension | 128 |
| Structural Decoder | Hidden dimension | Hidden dimension | 256 |
| | Heads | Number of heads in the transformer | 8 |
| | Blocks | Number of transformer layers | 4 |

Table 5: Training parameters for PhysFlow Model.

| Parameter | Value |
|---|---|
| Optimizer | Adam |
| Learning Rate | 0.0001 |
| $\beta_1, \beta_2, \epsilon$ | 0.9, 0.999, 1e-8 |
| Effective $M$ (max squared residues per batch) | 500k |
| Sequence masking probability | 50% |
| Minimum number of residues | 60 |
| Maximum number of residues | 512 |

## C ANGULAR AND FRAME REPRESENTATION

### C.1 ANGULAR REPRESENTATION

A protein backbone structure can be fully described using nine parameters per residue: three bond lengths, three bond angles, and three dihedral torsion angles. Under the common assumption of idealized bond lengths, only six angular parameters remain necessary to represent the internal structure. These angles are summarized in Table 6. Together, the nine parameters provide a lossless mapping between Cartesian and internal (angular) representations of the protein backbone.

Recall that given four atoms $a, b, c, d$ with positions $\mathbf{x}_a, \mathbf{x}_b, \mathbf{x}_c, \mathbf{x}_d$, the dihedral angle is defined as

$$\text{Dihedral}(a, b, c, d) = \text{atan2}\left(\hat{b}_2 \cdot (\hat{n}_1 \times \hat{n}_2), \hat{n}_1 \cdot \hat{n}_2\right) \qquad (24)$$

where

$$\hat{b}_2 = \frac{\mathbf{x}_c - \mathbf{x}_b}{\|\mathbf{x}_c - \mathbf{x}_b\|}, \quad \hat{n}_1 = \frac{(\mathbf{x}_b - \mathbf{x}_a) \times (\mathbf{x}_c - \mathbf{x}_b)}{\|\mathbf{x}_b - \mathbf{x}_a\|\|\mathbf{x}_c - \mathbf{x}_b\|}, \quad \hat{n}_2 = \frac{(\mathbf{x}_c - \mathbf{x}_b) \times (\mathbf{x}_d - \mathbf{x}_c)}{\|\mathbf{x}_c - \mathbf{x}_b\|\|\mathbf{x}_d - \mathbf{x}_c\|} \qquad (25)$$

Similarly, given three atoms $a, b, c$, the bond angle is defined as

$$\text{Angle}(a, b, c) = \arccos\left(\frac{(\mathbf{x}_a - \mathbf{x}_b) \cdot (\mathbf{x}_c - \mathbf{x}_b)}{\|\mathbf{x}_a - \mathbf{x}_b\|\|\mathbf{x}_c - \mathbf{x}_b\|}\right) \qquad (26)$$

Table 6: Angular representation.

| Angle | Description |
|---|---|
| $\phi$ | Dihedral $C_i N_{i+1} C_{\alpha_{i+1}} C_{i+1}$ |
| $\psi$ | Dihedral $N_i C_{\alpha_i} C_i N_{i+1}$ |
| $\omega$ | Dihedral $C_{\alpha_i} C_i N_{i+1} C_{\alpha_{i+1}}$ |
| $\theta_1$ | Bond angle $\angle N_i C_{\alpha_i} C_i$ |
| $\theta_2$ | Bond angle $\angle C_{\alpha_i} C_i N_{i+1}$ |
| $\theta_3$ | Bond angle $\angle C_i N_{i+1} C_{\alpha_{i+1}}$ |

Using these definitions, the internal coordinates of the $i$-th residue are obtained as

$$\psi_i = \text{Dihedral}\big((\mathbf{x}_N)_i, (\mathbf{x}_{C_\alpha})_i, (\mathbf{x}_C)_i, (\mathbf{x}_N)_{i+1}\big) \tag{27}$$

$$\phi_i = \text{Dihedral}\big((\mathbf{x}_C)_i, (\mathbf{x}_N)_{i+1}, (\mathbf{x}_{C_\alpha})_{i+1}, (\mathbf{x}_C)_{i+1}\big) \tag{28}$$

$$\omega_i = \text{Dihedral}\big((\mathbf{x}_{C_\alpha})_i, (\mathbf{x}_C)_{i+1}, (\mathbf{x}_N)_{i+1}, (\mathbf{x}_{C_\alpha})_{i+1}\big) \tag{29}$$

$$(\theta_1)_i = \text{Angle}\big((\mathbf{x}_N)_i, (\mathbf{x}_{C_\alpha})_i, (\mathbf{x}_C)_i\big) \tag{30}$$

$$(\theta_2)_i = \text{Angle}\big((\mathbf{x}_{C_\alpha})_i, (\mathbf{x}_C)_i, (\mathbf{x}_N)_{i+1}\big) \tag{31}$$

$$(\theta_3)_i = \text{Angle}\big((\mathbf{x}_C)_i, (\mathbf{x}_N)_{i+1}, (\mathbf{x}_{C_\alpha})_{i+1}\big) \tag{32}$$

## C.2 Forward Unfolding process in Cartesian Coordinates

Given a forward-simulated angular trajectory $\mathbf{u}_{0:T}$ as described in Section 4.2, the corresponding Cartesian flow can be obtained via nerf as $\mathbf{x}_{0:T}, \mathbf{v}_{0:T} = \text{nerf}(\mathbf{u}_{0:T})$, where computing velocities requires a Jacobian-based mapping between angular and Cartesian domains.

Alternatively, one can first transform the angular trajectory $\mathbf{z}_{0:T}$ into Cartesian space as $\mathbf{x}_{0:T} = \text{nerf}(\mathbf{z}_{0:T})$, and then compute velocities $v_t$ using finite differences (forward or backward) or cubic splines[3] . Moreover, a flow can also be defined directly in Cartesian coordinates by transforming the unfolding equations through nerf, as discussed above.

We can also derive an approximate equivalent framework for unfolding in Cartesian coordinates (centered by the geometric mean to ensure translation invariance), as described in Section 4.2. These unfolding equations operate directly in Cartesian space, incorporating Coulombic repulsion and an attractive potential toward the target state. This formulation provides greater flexibility in simulating the forward process.

Formally, the process can be expressed as a second-order Decay–Hamiltonian system (Neary and Topcu, 2023; Hamilton, 1834) evolving over Cartesian space:

$$\text{unfolding:} \quad \begin{cases} \dot{\mathbf{x}}_t & = \mathbf{v}_t \\ \dot{\mathbf{v}}_t & = f_{\text{unfold}}(\mathbf{x}_t, \mathbf{v}_t) = \underbrace{-\nabla_{\mathbf{x}_t} U(\mathbf{x}_t)}_{\text{potential force}} - \underbrace{K(\mathbf{v}_t)}_{\text{drag force}}, \end{cases} \tag{33}$$

where $U(\mathbf{x}_t)$ corresponds to an Ornstein–Uhlenbeck energy potential, and $K(\mathbf{v}_t)$ denotes a frictional drag term. We can also write the combined potential in terms of Cartesian coordinates as,

$$U(\mathbf{x}_t) = k_1 U_{\text{target}}(\mathbf{x}_t) + k_2 U_{\text{repulsion}}(\mathbf{x}_t) \tag{34}$$

that incorporates both Ornstein-Uhlenbeck attraction to $\mathbf{x}_{\text{target}}$ and Coulomb-like repulsion

$$U_{\text{target}}(\mathbf{x}_t) = \frac{1}{2} \sum_i \big(\mathbf{x}_{i,t} - \mathbf{x}_{i,\text{target}}\big)^2, \quad U_{\text{repulsion}}(\mathbf{x}_t) = \frac{1}{2} \sum_{i \neq j} \frac{1}{||\mathbf{x}_{i,t} - \mathbf{x}_{j,t}||}, \tag{35}$$

Here, $i, j$ denotes the index of backbone residues, $\mathbf{x}_{\text{target}} \sim p_0\big(\text{nerf}(\mathbf{z}_0)\big)$ with $\mathbf{z}_0 \sim p_0(\mathbf{z})$, and $k_1, k_2$ are potential weights. The target energy is a quadratic potential attracting the protein to the

---

[3] https://github.com/patrick-kidger/torchcubicspline

target state, while Coulombic repulsion (based on Euclidean distance) prevents collisions. Gradients $\nabla_{\mathbf{x}_t} U(\mathbf{x}_t)$ are computed via JAX autodiff.

To curb the oscillatory behavior from the potential, we utilize a linear drag force, $K(\mathbf{v}_t) = -\gamma \mathbf{v}_t$, where $\gamma > 0$ is the drag coefficient. This damping term counteracts large accelerations and stabilizes the dynamics, ensuring convergence to $\mathbf{x}_{\text{target}}$ as $t \to \infty$.

We parametrize the final target state to be a secondary structure linear chain predefined as linear $\beta$-sheets. Denoting $\mathbf{u} = [\mathbf{x}, \mathbf{v}]$, the terminal distribution is defined as:

$$p_0(\mathbf{u}) = \underbrace{\mathcal{N}(\mathbf{x}|\boldsymbol{\mu}_{\beta_x}, \sigma_{\beta_x}^2 I)}_{\text{position } p_0(\mathbf{z})} \cdot \underbrace{\mathcal{N}(\mathbf{v}|\mathbf{0}, \sigma_v^2 I)}_{\text{velocity}}, \tag{36}$$

where $\boldsymbol{\mu}_{\beta_x} = \text{nerf}(\boldsymbol{\mu}_\beta)$, with a small variation $\sigma_{\beta_x}^2$, around the coordinates computed from the $\beta$-sheet angles $\mu_\beta$. We also assume a data distribution

$$p_1(\mathbf{u}) = p_{\text{data}}(\mathbf{x}) \cdot \mathcal{N}(\mathbf{v}|\mathbf{0}, \sigma_v^2 I), \tag{37}$$

from which we have observations without velocities, and we augment the data with Gaussian velocities of variance $\sigma_v^2$.

Now we can follow the same methodology to compute forward simulations as described in Section 4.2 as,

$$p(\mathbf{u}_t|\mathbf{u}_1, \mathbf{u}_0) = \mathcal{N}\left(\begin{bmatrix} \mathbf{x}_t \\ \mathbf{v}_t \end{bmatrix} ; \begin{bmatrix} \mu_{\mathbf{x}_t} \\ \mu_{\mathbf{v}_t} \end{bmatrix}, \begin{bmatrix} \sigma_x^2 \\ \sigma_v^2 \end{bmatrix} I\right), \quad \begin{bmatrix} \mu_{\mathbf{x}_t} \\ \mu_{\mathbf{v}_t} \end{bmatrix} = \begin{bmatrix} \mathbf{x}_1 \\ \mathbf{v}_1 \end{bmatrix} - \int_t^1 \begin{bmatrix} \dot{\mathbf{x}}_t \\ \dot{\mathbf{v}}_t \end{bmatrix} d\tau \tag{38}$$

which provides us with the evolution of Cartesian coordinates and the corresponding velocities, which can be used to train the model with conditional flow matching. Below, describes a code snippet for the simulation,

```python
import numpy as np
import jax
import jax.numpy as jnp
from jax.tree_util import Partial
from functools import partial

def cdist2(X):
  r2 = jnp.sum( (X[:,None,:] - X[None,:,:])**2, axis=-1)
  return r2

def cdist(X):
  r2 = cdist2(X)
  I = jnp.eye(X.shape[0])
  r = jnp.sqrt(r2 + I) - I  # avoid sqrt(0) at diagonal, leads to grad=
    nan
  return r

def energy_attractor( theta,final_state,k=1 ):
    ener = jnp.sum((theta  - final_state)**2)
    return ener

def energy_coulomb( X ):
    N = X.shape[0]
    ids = 1 / ( cdist(X) + jnp.eye(N) )
    ener = jnp.sum(ids.sort(axis=-1)[:,:])
    return ener

@jax.jit
def energy_total(X, final_state,k1=1,k2=1):
    e_angle = energy_attractor(X,final_state)
    e_repulsion = energy_coulomb(X)
    return k1*e_angle + k2*e_repulsion

### Exemplar usage to simulate for a protein
```

```
34 cart_coords = X          #protein cartesian coords (e.g. C_alpha coordinates)
35 target = X_target        #target cartesian coords
36 Nt = 100                 #Number of time-steps for forward simulation
37 dt = 1/Nt
38 drag_const = gamma       #Drag force gamma constant
39 k1 = k1                  #weighting for attractor force
40 k2 = k2                  #weighting for repulsive force
41 #JAX Autodiff to compute grad
42 du = jax.grad( partial(energy_total, final_state=X_target,k1=k1,k2=k2) )
43 #Initialization for the system
44 H_cart = np.zeros((Nt,X.shape[0],X.shape[1]))
45 H_cart[0] = X
46 H_vel = np.random.normal(H_cart) #normal intialization or zero
       initialization
47
48 for i in range(1, Nt):
49     H_cart[i] = H_cart[i-1] + H_vel[i-1] * dt
50     force = du(H_cart[i-1])
51     H_vel[i] = H_vel[i-1] - force * dt - drag_const*H_vel[i-1]* dt
```

## C.3    PROOF OF PROPOSITION 1

To prove the statement, it suffices to show that under the translation of the Cartesian coordinates of a protein, the angular representation remains invariant.

Let $\mathbf{x} \in \mathbb{R}^{3N}$ denote the Cartesian coordinates of a protein with $N$ residues, and let $\mathbf{z}(\mathbf{x})$ denote its angular representation. For the $i$-th residue, we denote

$$\mathbf{x}_i = [(\mathbf{x}_N)_i, (\mathbf{x}_C)_i, (\mathbf{x}_{C_\alpha})_i]$$

as the Cartesian coordinates of the backbone atoms (Nitrogen, Carbon, and $\alpha$-Carbon, respectively). The corresponding angular representation is given as

$$\mathbf{z}(\mathbf{x}_i) = \left[\phi_i, \psi_i, \omega_i, (\theta_1)_i, (\theta_2)_i, (\theta_3)_i\right] \tag{39}$$

where $\phi_i, \psi_i, \omega_i$ are the backbone torsion angles and $(\theta_1)_i, (\theta_2)_i, (\theta_3)_i$ are the bond angles. Under the action of the translation operator $\mathcal{T}_\mathbf{t}$ on $\mathbf{x}_i$, the translated Cartesian coordinates can be represented as $\mathbf{x}_i' = \mathbf{x}_i + \mathbf{t}$, i.e.,

$$\mathbf{x}_i' = [(\mathbf{x}'_N)_i, (\mathbf{x}'_C)_i, (\mathbf{x}'_{C_\alpha})_i] \tag{40}$$

Recall the definitions of the bond angle and dihedral angle in equation 26 and equation 24. Both are computed from relative distances, which remain invariant under global translations of the Cartesian coordinates. For instance,

$$(\mathbf{x}'_N)_i - (\mathbf{x}'_C)_i = (\mathbf{x}_N)_i - (\mathbf{x}_C)_i, \quad (\mathbf{x}'_{C_\alpha})_i - (\mathbf{x}'_C)_i = (\mathbf{x}_{C_\alpha})_i - (\mathbf{x}_C)_i \tag{41}$$

which implies that the bond and dihedral angles remain unchanged under translation. Formally, this leads to having the same angular representation for both translated and original coordinates.

$$\mathbf{z}(\mathbf{x}_i) = \mathbf{z}(\mathbf{x}_i') \tag{42}$$

## C.4    FRAME REPRESENTATION

We continue from section 2 to describe the frame representation used by AF2 derived from Engh and Huber (2006) in more detail and follow the same definitions as described in Yim et al. (2023b). As discussed, $N^*, C_\alpha^*, C^*, O^*$ represents the idealized atom coordinates that assume chemically ideal bond lengths and angles. For unconditional generation, we take the idealized values of Alanine which are,

$$N^* = (-0.525, 1.363, 0.0) \tag{43}$$

$$C_\alpha^* = (0.0, 0.0, 0.0) \tag{44}$$

$$C^* = (1.526, 0.0, 0.0) \tag{45}$$

$$O^* = (0.627, 1.062, 0.0) \tag{46}$$

The backbone oxygen atom requires a torsional angle $\varphi_n$ to determine its coordinates:

$$O_n = T_n \cdot T(\varphi_n) \cdot O^*, \tag{47}$$

where $\varphi_n$ is the torsional angle for residue $n$ and

$$T^*(\varphi_n) = (\mathbf{r}_x(\varphi_n), \mathbf{x}_\varphi) \tag{48}$$

is the transformation frame following the methodology described in (Yim et al., 2023b). Here, $\varphi_n$ is a tuple specifying a point on the unit circle, $\varphi_n = [\varphi_{n,1}, \varphi_{n,2}]$, and

$$\mathbf{r}_x(\varphi_n) = \begin{bmatrix} 1 & 0 & 0 \\ 0 & \varphi_{n,1} & -\varphi_{n,2} \\ 0 & \varphi_{n,2} & \varphi_{n,1} \end{bmatrix}, \quad \mathbf{x}_\varphi = (1.526, 0, 0). \tag{49}$$

Thus, mapping from frames to atomic coordinates can be expressed as a single transformation as,

$$[N_n, (C_\alpha)_n, C_n, O_n] = \zeta_{\text{frame-to-atom}}(T_n, \varphi_n), \tag{50}$$

where $\zeta_{\text{frame-to-atom}}$ represents the full frame-to-atom transformation.

Now, we describe how to construct frames from coordinates by utilizing rigidFrom3Point algorithm in AF2 as,

$$\begin{aligned}
v_1 &= C_n - (C_\alpha)_n, \quad v_2 = N_n - (C_\alpha)_n \\
e_1 &= v_1/\|v_2\|, \quad u_2 = v_2 - e_1(e_1^\top v_2) \\
e_2 &= u_2/\|u_2\|, \quad e_2 = e_1 \times e_2 \\
\mathbf{r}_n &= \text{concat}(e_1, e_2, e_3) \\
\mathbf{x}_n &= (C_\alpha)_n \\
T_n &= (\mathbf{r}_n, \mathbf{x}_n)
\end{aligned}$$

where the initial computations are based on Gram-Schmidt. This whole procedure can be expressed as a single transformation as,

$$T_n = \zeta_{\text{atom-to-frame}}(N_n, (C_\alpha)_n, C_n). \tag{51}$$

