# OpenReview forum: "Let Physics Guide Your Protein Flows: Topology-aware Unfolding and Generation"
_ICLR.cc/2026/Conference — Submitted to ICLR 2026_

### Official Review · Reviewer_2ycY · 2025-10-16

**Soundness:** 3
**Presentation:** 1
**Contribution:** 2
**Rating:** 2
**Confidence:** 4

**Summary:**

The paper introduces PhysFlow, a protein–backbone generative model. Its key idea is a physics-inspired, nonlinear “noising” process that mimics protein unfolding. This process is formulated as a second-order decay–Hamiltonian system designed to prevent steric clashes during generation. The authors couple this physical process with flow matching on SE(3) generation. Experiments show that PhysFlow achieves excellent performance on both unconditional protein generation and sequence-conditioned monomer folding.

**Strengths:**

1. The work successfully adapts second-order dynamics to the complex geometry of protein structures, demonstrating a promising application of physical priors to protein deep learning domain. It can effectively reduce the clashes and atomic collisions.

2. The paper demonstrates compelling results across key benchmarks, especially in unconditional generation.

**Weaknesses:**

1. The most confusing aspect is the coordinate story in Section 4.2. Proposition 1 argues invariance in an angular representation and the forward ODE is defined in the latent $z$ (angle) space, yet the loss is pushed—via the chain rule—into Cartesian $x$-space, and the network architecture (e.g., IPA), loss function, part of energy also operate on $x$. What, then, is the precise role of the angular space? It seems that angular space is just used to introduce the Proposition 1, which is somehow meaningless (or can be substituted by $x$).
2. Angles and Cartesian coordinates are not globally bijective. They are only almost-everywhere invertible under restrictive conditions. In high-dimensional, complex systems it is easy to encounter angle configurations that do not map back uniquely to coordinates, errors accumulate through the transformation, and angles cannot encode bond lengths,  etc. How does the training ensure unbiased/accurate conversion between the two transformation?
3. The manuscript lacks an ablation to justify the necessity of second-order dynamics. While a second-order decay–Hamiltonian formulation is intuitively reasonable, the paper does not demonstrate that it is required. For example, would a first-order ODE with explicit collision-avoidance terms perform worse? Does the second-order just aim for novelty?
4. Typically, joint training of conditional and unconditional objectives slightly degrades folding performance—especially in the conditional setting. Could you report results for the conditional folding task on CAMEO2022 (e.g., TM-score, RMSD, GDT-TS, IDDT), and include stability/geometry metrics that may highlight PhysFlow’s advantages (e.g., Cα clash rate, peptide-bond breaks)?
5. Sections 1 and 2 are weakly written. The introduction lacks a broad discussion of the protein folding task and its relevance to drug design. A second-order Hamiltonian/Langevin formulation has been widely explored in diffusion and flow frameworks (underdamped/kinetic samplers, second-order flow matching, acceleration with momentum, etc.). The authors miss discussion about this content. By contrast, much of Section 2 and Section 4.1 feels peripheral to the main contribution.

**Questions:**

The same as Weaknesses. If the authors can address these concerns, I would be inclined to increase my score.

---

> ### Author Response · Authors · 2025-11-20
> **Rebuttal**
>
> Thank you for your feedback! We address all your questions and comments below.
>
> > The most confusing aspect is the coordinate story in Section 4.2....
>
> Thank you for the opportunity to clarify this.
>
> - **Regarding choice of coordinate system**: The angular space defines the unfolding process in a way that preserves the structural integrity of the protein chain and prevents self-collisions. Operating in angular coordinates inherently enforces idealized bond lengths and local connectivity between residues—constraints that are not guaranteed when the flow is defined directly in Cartesian space. However, since the angular representation alone cannot fully capture inter-residue collisions, it is intermittently converted to Cartesian coordinates to evaluate the repulsion energy, which forms a component of our augmented potential. For completeness, we also describe an analogous flow in Cartesian space in Appendix C.2, though this formulation does not strictly maintain the bond-length constraints enforced in angular space.
>
> - **Regarding learning the reverse**: Empirically, neural architectures such as IPA have been shown to perform better when operating in Cartesian coordinates [1,2], as they provide greater flexibility in modeling side-chain orientations and other local structural variations that are not easily captured in angular coordinates. To leverage the strengths of both representations, we define the forward unfolding process in angular space—where structural constraints are naturally encoded—but train the model in Cartesian space, which facilitates more expressive and stable learning. This hybrid approach allows us to retain geometric consistency while improving the modeling capacity of the learned reverse process.
>
>
> [1] SE(3) diffusion model with application to protein backbone generation, ICML 2023
>
> [2] Protein structure generation via folding diffusion, Nature
>
> > Angles and Cartesian coordinates are not globally bijective.....
>
> Thank you for raising this point.
>
> ### Bijection between Cartesian and Angular Representation
> We agree that angular and Cartesian representations are not globally bijective. However, in practice, protein structures occupy a restricted, physically valid manifold in which the mapping is well-behaved. In our framework, the conversion from angular to Cartesian coordinates assumes idealized bond lengths—the same assumptions made in frame representation—and utilizes the NeRF algorithm for reconstruction from dihedral and bond angles.
>
>
> In response to your remark, we evaluated the reconstruction fidelity of this NeRF-based transformation [1]. Specifically, we converted native structures into angular coordinates and then reconstructed them in  Cartesian space under different assumptions (idealized lengths only vs. idealized lengths + bond angles). We report the average TM-scores between the ground-truth and reconstructed structures below:
>
> | Method                                      | Reconstructed TM Score |
> |---------------------------------------------|--------------------------|
> | No idealized                                | 1.00                     |
> | Idealized bond lengths                      | 0.92                     |
> | Idealized bond lengths and bond angles      | 0.52                     |
>
> These results show that when true bond lengths, dihedrals, and angles are used, the reconstruction is exact (TM = 1.00). Even when only idealized bond lengths are applied, the reconstruction remains high-fidelity (TM = 0.92). The degradation only appears when both bond lengths and angles are idealized simultaneously, which represents a highly simplified and unrealistic setting.
>
>
> [1] NeRFax: An efficient and scalable conversion from the internal representation to Cartesian space

---

> > ### Author Response · Authors · 2025-11-20
> > **Rebuttal**
> >
> > > Typically, joint training of conditional and unconditional objectives slightly degrades...
> >
> > Thank you for the suggestion. We report results on the CAMEO2022 dataset [1] below, including both folding accuracy metrics and structural stability/geometry metrics:
> >
> > | Method                | TM Score | RMSD  | GDT-TS | IDDT | CA clash (%) | PepBond break (%) |
> > |-----------------------|----------|-------|--------|------|--------------|--------------------|
> > | EigenFold†            | 0.74     | 7.65  | 0.70   | 0.73 | 8.0          | N/A                |
> > | ESMFold†              | 0.84     | 3.98  | 0.82   | 0.87 | 0.3          | 4.7                |
> > | RoseTTAFold2†         | 0.86     | 3.52  | 0.84   | 0.89 | 0.3          | 5.5                |
> > | OpenFold†             | 0.87     | 3.21  | 0.85   | 0.89 | 0.4          | 2.0                |
> > | AlphaFold 2†          | 0.87     | 3.21  | 0.86   | 0.90 | 0.3          | 4.8                |
> > | FoldFlow 2        | 0.55     | 14.21 | 0.55   | 0.53 | 0.9          | 4.7                |
> > | PhysFlow          | 0.60     | 12.11 | 0.60   | 0.58 | 0.2          | 2.1                |
> >
> > We observe that PhysFlow outperforms FoldFlow-2, which is trained on the same monomeric dataset, achieving significantly lower CA clash rates and PepBond breaks. This demonstrates that PhysFlow’s physics-informed unfolding and denoising process improves structural plausibility. We note that methods such as ESMFold, AlphaFold 2, and other large-scale models are trained on much larger and more diverse datasets; therefore, direct comparisons to these models are not fully fair.
> >
> > [1] ProteinBench: A Holistic Evaluation of Protein Foundation Models
> >
> > > The manuscript lacks an ablation to justify the necessity of second-order dynamics...
> >
> > Thank you for raising this point. Our use of second-order dynamics stems from the physical formulation of the forward unfolding process. In particular:
> >
> > - **Potential-based forces inherently induce second-order dynamics.** Our forward process is governed by an augmented energy potential consisting of both attractive and repulsive components. The gradient of this potential defines a force, driving the acceleration of $\dot{v}$. This yields a natural Hamiltonian system describing the evolution of the system. A first-order ODE cannot capture such force-driven motion without breaking the physical meaning of the potential.
> >
> > - Incorporating a repulsion drift directly into a first-order system would not correspond to any valid conservative potential field; therefore, it cannot recover valid Hamiltonian dynamics. This eliminates the connection between the potential, the induced forces, and the unfolding trajectory.
> >
> > - **First-order alternatives lose physical grounding**. While one could attempt to engineer first-order dynamics with heuristic repulsion terms, such a formulation would no longer correspond to motion under a potential driven by the force and would remove the physics-based motivation behind our unfolding process.
> >
> > > Sections 1 and 2 are weakly written...
> >
> > Thank you for raising this point. We will incorporate these broader discussions and improve Sections 1 and 2 in the revised version of the paper.
> >
> > We are grateful for your thoughtful comments and suggestions, which have allowed us to emphasize some salient aspects, and shed light on subtle facets of the proposed method. We hope that our response addresses your concerns and reinforces your support for this work.

---

### Official Review · Reviewer_uDYi · 2025-10-27

**Soundness:** 1
**Presentation:** 2
**Contribution:** 1
**Rating:** 2
**Confidence:** 5

**Summary:**

The paper introduces PhysFlow, a generative model for protein backbone structures. The authors' core motivation is that standard diffusion and flow models use a "noising" process that is not physically realistic, ignoring principles like topological integrity and steric clashes. To address this, they propose a physics-inspired, non-linear "unfolding" process grounded in Hamiltonian dynamics. This forward process is designed to unfold a protein into a secondary structure (like a $\beta$-sheet) while explicitly preserving bonds and using a Coulomb-like repulsion term to prevent residue collisions.This physics-driven process is integrated into the flow-matching paradigm on $SE(3)$ to learn the distribution of protein backbones. The model also incorporates sequence information, enabling it to perform both unconditional backbone generation and sequence-conditioned folding. The authors claim that PhysFlow achieves state-of-the-art performance in unconditional generation, producing more designable and novel structures, and accurately folds monomer sequences.

**Strengths:**

The paper's motivation to incorporate more physics into the generative forward process is a novel and interesting research direction. The idea of replacing a generic noise process with a more structured unfolding trajectory that respects physical constraints like collision avoidance is intuitive.

**Weaknesses:**

1. The paper proposes a complex, non-linear, second-order Hamiltonian system for the forward unfolding process but completely fails to discuss or justify the reverse generative process. It is a major theoretical gap to assume that a standard, first-order flow-matching objective is sufficient to learn the true time-reversal of such a complex non-linear dynamic system, which is required to satisfy the Continuity Equation or the Fokker-Planck equation. The original reverse processes for diffusion models or flow matching generally cannot be directly applied here without theoretical modification.
2. The model is explicitly designed to unfold to a target distribution $p_0$ defined as a linear $\beta$-sheet, yet the empirically generated samples are biased toward the $\alpha$-helix structure (82.2%). This contradiction fundamentally weakens the paper's core claim, suggesting that the physics-guided forward process is either not being reversed correctly or has little impact on the final generated distribution.
3. The paper's motivation, that a "physically plausible" forward trajectory is necessary, is questionable. The power of modern generative models originates from the strong generalization ability of the denoising network, which requires the forward process to be stochastic and highly noisy to enable learning recovery from arbitrary, highly corrupted states. Restricting the model to a single, well-defined forward path may hurt its ability to generalize to novel states not encountered on this narrow trajectory.
4. The proposed algorithm requires an expensive pre-processing step to simulate and store the forward trajectory for every protein. This prevents on-the-fly training and introduces a critical scalability bottleneck, especially since the forward simulation takes over one second per protein (as shown in Figure 4), severely limiting its application to larger datasets like AlphaFold DB.
5. The evaluation is undermined by non-standard and weak comparisons. (1) The authors compare against some models trained on shorter sequences (e.g., up to 256 aa) while generating samples up to 300 aa, leading to an unfair comparison. (2) They exclude stronger baselines (like Proteina and Genie2) with the excuse of using different datasets, even though this paper uses a self-curated dataset as well. (3) The comparison for the folding task (Table 3) is made against co-design models (MultiFlow, FoldFlow-2) rather than specialized, state-of-the-art folding models, making the claim of strong folding performance unconvincing.
6. The reported "Diversity Cluster Ratio" is affected by the designability value. For a clear assessment of structural variety, the authors should report the absolute number of designable clusters instead, which is a more robust metric for diversity.
7. The paper lacks definitions for key terms, specifically "linear diffusion" in the first paragraph of Section 3 and "unfolding flow" (Line 187).
8. The choice to set the target state $p_0$ from a beta-sheet distribution is arbitrary and unexplained. Ideally, the model should be able to generate any secondary structure, so the selection of a single predefined distribution for the "unfolded" state needs strong justification.
9. The paper is lacking crucial hyperparameter configurations for the forward simulation, including $\gamma$ (drag coefficient), $\sigma_\beta^2$, $\mu_\beta$ (for the target distribution), and $\sigma_v^2$, hindering the reproducibility of the work.
10.  In Table 1, the categorization of existing models into DDPM, CFM, OT, or DSM is an oversimplification, as these algorithms are well-known to be deeply related and can be unified under the Stochastic Interpolant framework.
11. Equation (3) is wrong, as in flow matching, there is only one ODE. The reverse process simply simulates the same ODE in the reverse time direction.

**Questions:**

See above.

---

> ### Author Response · Authors · 2025-11-20
> **Rebuttal**
>
> Thanks so much for your thoughtful comments and excellent suggestions! We've acted on all of them, and also address all your concerns, as we describe below.
>
> >  The paper proposes a complex, non-linear, second-order Hamiltonian system for the forward unfolding process...
>
> Thank you for the opportunity to clarify this point.
>
> The forward unfolding dynamics we introduce are formulated as a system of Hamiltonian ODEs [1,2]. Importantly, for such systems, the reverse-time generative process does not require explicitly solving a second-order reverse Hamiltonian system. Instead, as shown in [2], the reverse process can be modeled using a first-order velocity field that captures the probability flow induced by the Hamiltonian dynamics.
>
> In particular, the model only needs to learn the velocity field associated with the forward trajectories. This velocity predictor defines the probability paths in reverse time and is theoretically justified under the condition that the learned vector field is divergence-free, which ensures consistency with the deterministic Fokker–Planck equation (see Appendix H in [2]). Under this condition, a standard flow-matching objective is sufficient to recover the correct reverse-time dynamics, even when a more complex Hamiltonian system governs the forward process.
>
> We will include this clarification in the revised version of the paper.
>
>
> [1] MCMC using Hamiltonian dynamics
>
> [2] Hamiltonian Score Matching and Generative Flows, NeurIPS 2024
>
> > The model is explicitly designed to unfold to a target distribution $p_0$ defined as a linear $\beta$-sheet,...
>
> Thank you for the observation. The forward noising (unfolding) process is designed to destroy structural information by transforming the protein into a non-informative prior, modeled as a linear $\beta$-sheet (i.e., a linear chain of amino acids). The reverse process is then learned to invert this transformation.
>
> It is a well-established fact [1,2] that protein generative models are more prone to produce $\alpha$-helical substructures, as these are generally more designable than other secondary structures. This bias is further reinforced by annealing strategies commonly used in the reverse process to guide generation toward more designable structures [1,3,4]. Our work focuses on learning a physics-guided forward process rather than studying the reasons for the $\alpha$-helix bias, which is a property shared by most generative models, including ours.
>
> Although we define the target prior as a linear $\beta$-sheet, there is no restriction on the choice of prior distribution, and the model remains capable of generating a variety of secondary structures.
>
> [1] Proteina: Scaling Flow-based Protein Structure Generative Models, ICLR 2025
>
> [2] Assessing Generative Model Coverage of Protein Structures with SHAPES, Cell Systems
>
> [3] SE(3)-Stochastic Flow Matching for Protein Backbone Generation, ICLR 2024
>
> [4] Sequence-Augmented SE(3)-Flow Matching For Conditional Protein Backbone Generation, NeurIPS 2024
>
>
> > The paper's motivation, that a "physically plausible" forward trajectory is necessary, is questionable....
>
> Thank you for raising this point. Although the unfolding dynamics follow a physically motivated Hamiltonian formulation, stochasticity is injected at multiple stages of the process. Specifically:
>
> - **Initial state variability**: The prior state is sampled from a normal distribution with non-zero variance, ensuring that the system begins from diverse unfolded configurations rather than a single deterministic state. Thus, no two forward trajectories are identical.
>
> - As described in Eq. 12, intermediate states are also sampled with non-zero variance, which exposes the denoising network to a broad distribution of corrupted, partially unfolded, and transitional configurations.
>
> As a result, the denoiser does not learn to invert a narrow, single-path trajectory. Instead, it learns to recover from a continuum of noisy, perturbed states, analogous in spirit to classical diffusion/flow models but grounded in physically plausible transitions.
>
> > The proposed algorithm requires an expensive pre-processing step..
>
> Thank you for raising this point. We respectfully disagree with the notion that the forward simulation constitutes a major scalability bottleneck. While our method indeed requires generating forward simulation trajectories, this process is embarrassingly parallelizable and can be efficiently distributed across multiple CPU cores or nodes. The runtime reported in Figure 4 reflects sequential processing for clarity of presentation; however, in practice, forward simulations for thousands of proteins can be executed in parallel, reducing the effective wall-clock time per protein to well below one second. Thus, this step does not pose a practical limitation for scaling to larger datasets such as AlphaFold DB.

---

> > ### Author Response · Authors · 2025-11-20
> > **Rebuttal**
> >
> > > The evaluation is undermined by non-standard and weak comparisons....
> >
> > Thank you for the opportunity to clarify this aspect.
> >
> > - **Sequence length and fairness of comparison**:  We followed the evaluation protocol described in FoldFlow [1], which is also used by  baseline methods, including FrameDiff [2], FoldFlow-2 [3], and FrameFlow [4]. Specifically, FoldFlow (OT/CFM) was trained on monomer datasets from the PDB with sequence lengths ranging from 60 to 512 residues (see Appendix I.5 in [1]). Hence, sequences of length 300 fall well within the training distribution. Moreover, their evaluation explicitly included protein lengths in $\{100, 150, 200, 250, 300\}$ (see Section 5.2 in [1]). We use the Proteina [5] model, which in their original work is also used to generate and evaluate longer protein sequences. RFDiffusion [6] is trained on the same data used for RoseTTAFold [7] and is additionally finetuned on sequences up to length 384. Chroma [8] is trained on sequences ranging from 50 to 1000 residues, with evaluation performed across the same range.
> >
> > Because these methods differ in training datasets and sequence-length cutoffs, we use a dagger notation to clearly indicate cases where a method was not trained on exactly the same sequence lengths. Among the baselines, Genie is the only method trained up to 256 residues; we mark this with a dagger as well. We include Genie in our comparison for completeness, following prior baseline works [1,3,4].
> >
> > To ensure a fair and consistent comparison, we evaluate all methods using the sequence lengths employed in the original protocols.
> >
> > In response to your suggestion, we have additionally evaluated proteins with lengths in the range 50–250 residues. We also compared our method to RFDiffusion without pretraining, as described in [2]. The results for designability, novelty, and diversity are summarized in table below,
> >
> > | Method                       | Designability | Novelty (TM-Frac) | Diversity (Cluster) |
> > |------------------------------|---------------|--------------------|-----------------------|
> > | RFDiffusion†                 | 0.64          | 0.40               | 0.45    (217)               |
> > | Proteina†                    | 0.98          | 0.38               | 0.30      (150)            |
> > | RFDiffusion (no-pretraining) | 0.24      | 0.36               | 0.31      (145)            |
> > | FrameDiff                    | 0.64          | 0.52               | 0.38             (126)     |
> > | FrameFlow                    | 0.88          | 0.47               | 0.51             (236)     |
> > | FoldFlow                     | 0.96          | 0.44               | 0.38               (170)   |
> > | PhysFlow                     | 0.97          | 0.44               | 0.43               (220)   |
> >
> > We observe that RFDiffusion can generate designable sequences only when initialized with pre-trained weights [2]. By contrast, PhysFlow achieves consistently high designability while maintaining competitive novelty and diversity across this broader range of sequence lengths. These results highlight the robustness of PhysFlow in generating structurally and functionally plausible proteins without relying on pretraining.
> >
> > - **Choice of baselines**: For the sequence-conditioned monomer folding task, we use a self-curated dataset solely for evaluation purposes. All baseline methods (FoldFlow-2 and MultiFlow) are trained and evaluated on the same training and test sets, ensuring a fair comparison. On unconditional generation models like Proteina and RFDiffusion are excluded because they are trained on much larger datasets, which prevents a controlled, apples-to-apples comparison. Including them would risk introducing dataset biases that could confound the evaluation.
> >
> > - **Folding task comparisons**: Our goal in Table 3 is to evaluate sequence-conditioned co-design approaches for monomer folding, which is why we compare against co-design models like MultiFlow and FoldFlow-2. These baselines represent the state-of-the-art for the co-design setting. Comparing against general-purpose folding models (e.g., AlphaFold variants) is not directly comparable, as these models are trained on much larger datasets. Retraining under these conditions would require substantial computational resources, which are not easily available. Overall, our evaluation follows standard and widely adopted protocols, ensuring fair and meaningful comparisons across methods.

---

> > > ### Author Response · Authors · 2025-11-20
> > > **Rebuttal**
> > >
> > > [1] SE(3)-Stochastic Flow Matching for Protein Backbone Generation, ICLR 2024
> > >
> > > [2] SE(3) diffusion model with application to protein backbone generation, ICML 2023
> > >
> > > [3] Sequence-Augmented SE(3)-Flow Matching For Conditional Protein Backbone Generation, NeurIPS 2024
> > >
> > > [4] Fast protein backbone generation with SE(3) flow matching, preprint
> > >
> > > [5] Proteina: Scaling Flow-based Protein Structure Generative Models, ICLR 2025
> > >
> > > [6] De novo design of protein structure and function with RFdiffusion, Nature
> > >
> > > [7] Accurate prediction of protein structures and interactions using a three-track neural network, Science
> > >
> > > [8] Illuminating protein space with a programmable generative model, Nature
> > >
> > > > The paper lacks definitions for key terms, specifically "linear diffusion" in the first paragraph of Section 3 and "unfolding flow" (Line 187).
> > >
> > > In the paper, “linear diffusion” refers to the class of forward processes commonly used in diffusion models, such as the Ornstein–Uhlenbeck (OU) or mean-reverting processes, where noise is added in a linear and coordinate-independent manner across time steps. In contrast, the “unfolding flow” denotes the forward process we introduce based on Hamiltonain dynamics.
> > >
> > > We will add precise definitions and a brief conceptual explanation of both terms in the revised version of the paper to ensure clarity.
> > >
> > > > The choice to set the target state $p_0$  from a beta-sheet distribution is arbitrary and unexplained.
> > >
> > > Thank you for the opportunity to clarify this aspect. Although we define the target state as a linear $\beta$-sheet, this choice does not restrict the model’s ability to generate other secondary structures, as the prior distribution can be defined arbitrarily. The linear $\beta$-sheet was chosen based on first-principles biological reasoning: an unfolded linear chain of amino acids serves as a physically meaningful prior, analogous to white noise in generative models, providing no specific structural bias. This ensures that the model learns the folding generative process from an unbiased initial state while remaining physically grounded.
> > >
> > > > The paper is lacking crucial hyperparameter configurations for the forward simulation,...
> > >
> > > We thank the reviewer for pointing this out. The hyperparameters used in the forward simulation are provided below. The angular coordinates are defined as $\mu_{\beta} = [\phi,\omega,\psi,\theta_1,\theta_2,\theta_3]$, which represent the standard values of the dihedral angles of the linear $\beta$ sheet in radians [1,2,3]. The drag coefficient is scaled with the length of the protein as longer proteins require more drag than shorter ones.
> > >
> > > | Hyper-parameter   | Value                                |
> > > |-------------------|---------------------------------------|
> > > | $\gamma$                 | $0.01\ell_{prot}$                            |
> > > | $\sigma^2_{\beta}$              | 0.1                                   |
> > > | $\mu_{\beta}$               | [2.26, 3.14, -2.44, 1.095, 0.98, 1.20] |
> > > | $\sigma^2_{v}$              | 0.1                                   |
> > >
> > > We will add them in the revised version of the paper.
> > >
> > > [1] https://en.wikipedia.org/wiki/Beta_sheet
> > >
> > > [2] Intrinsic $\alpha$‐helical and $\beta$‐sheet conformational preferences: A computational case study of alanine, Protein Science
> > >
> > > [3] [Beta-sheet angles](https://bio.libretexts.org/Bookshelves/Biochemistry/Fundamentals_of_Biochemistry_(Jakubowski_and_Flatt)/01%3A_Unit_I-_Structure_and_Catalysis/04%3A_The_Three-Dimensional_Structure_of_Proteins/4.02%3A_Secondary_Structure_and_Loops)
> > >
> > > > In Table 1, the categorization of existing models into DDPM, CFM, OT, or DSM is an oversimplification..
> > >
> > > We agree that all the methods can indeed be unified under the Stochastic Interpolant framework. In our paper, we opted to categorize them based on their modeling and noising process strategies for clarity and ease of comparison. We will add a note in the revised manuscript acknowledging the connection to the Stochastic Interpolant framework.
> > >
> > > > Equation (3) is wrong, as in flow matching, there is only one ODE...
> > >
> > > We thank the reviewer for the observation. Equation (3) corresponds to the standard formulation of continuous normalizing flows (CNFs) [1,2], where the forward and reverse dynamics are governed by the same vector field $\mathbf{f}_{\theta}$. The reverse process thus integrates the same ODE backward in time (from $t=1$ to $t=0$) by negating the sign of the time differential. We will revise the text to clarify that both directions correspond to a single underlying ODE whose trajectory can be simulated in either temporal direction.
> > >
> > > [1] Neural Ordinary Differential Equations, NeurIPS 2018
> > >
> > > [2] FFJORD: Free-form Continuous Dynamics for Scalable Reversible Generative Models, ICLR 2019
> > >
> > > Again, thank you so much for your extremely constructive feedback. We hope we have sufficiently addressed your concerns, and would appreciate your stronger support for this paper.

---

> > > > ### Comment · Reviewer_uDYi · 2025-11-26
> > > >
> > > > Thanks for the authors' detailed response, though most of my concerns remain.
> > > >
> > > > First, thanks for the authors' providing the reference to the paper "Hamiltonian Score Matching and Generative Flows" to justify the design of reverse process and training of velocity field. However, the proof here is non-trivial as discussed in that paper. Despite using the same framework, it is weird that the authors didn't cite that paper in the submission.
> > > >
> > > > Second, regarding the choice of beta-sheet distribution as $p_0$. The motivation of using beta-sheet distribution as $p_0$ contradicts with the results of overrepresentation of alpha-helix. This concern is still unexplained.
> > > >
> > > > Third, I find the authors wrongly cite the Proteina's results in comparison. (1) Proteina provides separate models for standard and long protein generation. For benchmarking protein generation with 300AA, the long protein generation model should be used. (2) The author selects the "worst" model results in Proteina paper to report ("worst" in terms of designable clusters). The "best" version of Proteina generates 323 designable clusters as reported in the original paper, which is signficantly higher than the proposed method.
> > > >
> > > > Overall, I still think there are many concerns about the motivation, soundness and experiments of the proposed method. Thus I keep my score unchanged.

---

> > > > > ### Author Response · Authors · 2025-12-01
> > > > > **Reply**
> > > > >
> > > > > Thank you again for your continued engagement. We appreciate the opportunity to clarify the remaining concerns.
> > > > >
> > > > > > Second, regarding the choice of beta-sheet distribution as....
> > > > >
> > > > > ### Linear beta sheet prior corresponds to a physical prior providing no bias in generative process
> > > > > On the choice of the linear beta-sheet prior. The use of a linear backbone as the prior is not intended to impose beta-sheet structural bias; rather, it reflects the biophysically accurate unfolded state of a polypeptide chain i.e. a linear chain of amino acids. This state is well-established in protein folding literature as a physically meaningful, unbiased starting point—directly analogous to white noise in generative modeling. Importantly, the extended chain does not encode any preference for beta sheets, alpha helices, or any other secondary structure. It reflects the absence of folded conformations.
> > > > >
> > > > > > Proteina comparisons.
> > > > >
> > > > > ### More diverse proteina model has worse designability
> > > > > As also noted in our rebuttal, we used the Proteina model fine-tuned specifically for longer sequences, which is the appropriate counterpart for 300-residue generation. Regarding the choice of results: we compared against the Proteina variant that achieves the highest designability (99%). The diversity numbers we reported correspond to this model. The alternative Proteina variant you reference—yielding 323 designable clusters—indeed exhibits higher diversity, but its designability drops to 91%, substantially lower than our method's 98%. The designability is the key important metric as if the designability is low the diversity becomes redundant. Our intention was to compare against the strongest Proteina baseline on the key criterion of designability, while also reporting the diversity associated with that same model.
> > > > >
> > > > > We hope these clarifications address your concerns and provide additional context for the methodological choices and comparative evaluations.

---

### Official Review · Reviewer_3bsB · 2025-10-31

**Soundness:** 1
**Presentation:** 2
**Contribution:** 1
**Rating:** 2
**Confidence:** 4

**Summary:**

The authors propose a new non-linear noising process for protein structures that is inspired by Hamiltonian dynamics that is supposed to reduce clashes and steric violations and train a structure generation model similar to FoldFlow with it. They compare it to other models on unconditional backbone design and sequence conditioned monomer folding.

**Strengths:**

[S1] Novel approach: the authors depart from the common linear noising schedule and propose an interesting physics-inspired noising process, and introduce modifications to their model training such as the Look-Ahead loss that help their model to learn this denoising objective (an ablation of this would add to the paper).

[S2] Comparing to models trained on the same dataset for structure generation, they have favorable performance in unconditional backbone design.

**Weaknesses:**

[W1] Theu authors claim that linear noising processes make “models prone to steric clashes and violations”. However, in previous work there has been extensive validation that models with linear noising processes and without any auxiliary losses have very strong biophysical plausibility with clashes and violations on the level of experimental structures (Geffner, Tomas, et al. "La-proteina: Atomistic protein generation via partially latent flow matching." arXiv preprint arXiv:2507.09466 (2025).). The authors do not substantiate their claim with evidence, I would therefore recommend to remove this claim which most of their motivation relies on; there is no evaluation of biophysical plausibility, steric clashes or violations in the paper. Therefore the motivation for the topology-aware unfolding process seems weak and its advantage not demonstrated. In addition, the authors themselves use auxiliary losses to supervise bond distances etc and avoid clashes; if the proposition of the authors would be correct their method should not need to rely on these auxiliary losses since other methods achieve high performance without them.

[W2] the authors claim that “PhysFlow achieves state-of-the-art performance in unconditional protein backbone generation and sequence-conditioned monomer folding”. Both of these statements are not true: in Table 2 PhysFlow is clearly worse than a number of methods, including RFDiffusion, Chroma and Proteina. The fact that these models are trained on different datasets is no excuse: the authors could themselves have trained their model on these openly available datasets. So claiming SOTA performance while ignoring all strong baselines is not justified. Similarly for folding performance: they show that their method achieves 11-15A folding performance, making it useless from a practical perspective. It is way worse than ESMFold, and again just because ESMFold was trained on a different dataset that does not justify claiming SOTA performance by ignoring that method.

**Questions:**

[Q1] In practice researcher use these models for conditional tasks such as binder design or moti scaffolding. Do you think your current physical noising process is suitable for these tasks or would you do additional adjustments to it to account for the different tasks?

[Q2] Many modern protein structure generation generate not only the backbone structure but all-atom structures including sequences. Can your framework be used in that respect, and if yes how?

---

> ### Author Response · Authors · 2025-11-20
> **Rebuttal**
>
> Many thanks for your thoughtful review. We address your concerns and incorporate your suggestions below.
>
> > Theu authors claim that linear noising processes make “models prone to steric clashes and violations”.
>
> Thank you for the opportunity to clarify this.
>
> - **Analysis of clashes**: We have quantified the average number of residue–residue steric collisions (defined as pairwise residue distances $< d(x_i,x_j)$(nm)) observed across the forward trajectories of proteins. This analysis was performed for Cartesian linear diffusion/flow baselines as well as for variants of our PhysFlow method with different weightings $(k_1, k_2)$ as shown in Fig. 5. We observe that the number of collisions decreases as the weight on Coulombic repulsion ($k_2$) increases within the PhysFlow framework, demonstrating that PhysFlow can effectively mitigate steric clashes compared to linear flow baselines.
>
>
> - **Auxiliary losses**: The auxiliary losses in PhysFlow are not primarily used to prevent clashes but also to refine local geometric accuracy—specifically to better capture fine-grained structural details that are not easily represented in purely angular coordinates. Similar auxiliary losses are also employed in recent backbone and atomistic generative models such as FrameDiff [1], FoldFlow-1 $\&$ 2 [2,3], FrameFlow [4], and Proteina [5].
>
> Regarding the reference to La Proteina, we note that this model inherits the same architectural design as Proteina, which is not strictly SE(3)-equivariant or invariant. Instead, it approximates rotational invariance by introducing random rotations during training—a heuristic whose effects have not been systematically studied. Moreover, Proteina and La Proteina use much larger datasets, making it unclear whether their reported performance gains arise from the model design itself or these auxiliary datasets and fine-tuning strategies. Therefore, direct comparisons between PhysFlow and such models are not strictly apples-to-apples, as PhysFlow explicitly enforces SE(3) symmetry through its topology-aware unfolding mechanism.
>
> [1] SE(3) diffusion model with application to protein backbone generation, ICML 2023
>
> [2] SE(3)-Stochastic Flow Matching for Protein Backbone Generation, ICLR 2024
>
> [3] Sequence-Augmented SE(3)-Flow Matching For Conditional Protein Backbone Generation, NeurIPS 2024
>
> [4] Fast protein backbone generation with SE(3) flow matching, preprint
>
> [5] Proteina: Scaling Flow-based Protein Structure Generative Models
>
> > the authors claim that “PhysFlow achieves state-of-the-art performance in unconditional protein backbone generation and sequence-conditioned monomer folding”...
>
> Thank you for the opportunity to clarify this aspect.
>
> - **Fairness of comparison** : We followed the evaluation protocol described in FoldFlow [1], which is also used by all baseline methods, including FrameDiff, FoldFlow-2, and FrameFlow. Stronger models like Proteina, Genie2, or RFDiffusion are excluded are trained on much larger and distinct datasets with more techniques such as LoRA, fine-tuning,etc. , which would prevent a controlled, \textit{apples-to-apples} comparison. While it is technically possible to retrain our model on these datasets, doing so would require extensive computational resources, which are not as readily available. Importantly, this does not affect the fairness of our comparisons with baselines trained on the same dataset.
>
> - **Folding task comparisons**: In Table 3, our goal is to evaluate sequence-conditioned co-design approaches for monomer folding. We therefore compare against co-design models like MultiFlow and FoldFlow-2 using the same training and testing datasets to provide a controlled and meaningful comparison. Comparing against general-purpose folding models (e.g., ESMFold, AlphaFold variants) is not directly comparable, as these models are trained on much larger datasets. Retraining under these conditions would require substantial computational resources, which are not easily available.
>
> Nonetheless, we appreciate your comment and will update this claim accordingly in the revised version of the paper.

---

> > ### Author Response · Authors · 2025-11-20
> > **Rebuttal**
> >
> > > In practice researcher use these models for conditional tasks such as binder design or moti scaffolding...
> >
> > Based on the your remark we have conducted a motif scaffolding experiment based on RFDiffusion benchmark [1].
> >
> > To enable PhysFlow to perform motif-scaffolding, we add two additional features to our model:
> >
> > -  The motif structure remains fixed (i.e., not unfolded), while only the remaining positions—determined by the motif mask—are unfolded during the forward simulation.
> >
> > - The fixed motif coordinates are also provided as conditional inputs to the model, supplying structural context throughout the denoising process.
> >
> > Here are the results on it compared to the baselines
> >
> > | Task Name              | Proteina† | Genie2† | RFDiffusion† | FrameFlow | PhysFlow |
> > |------------------------|-----------|---------|--------------|-----------|----------|
> > | #Successes total       | 2094      | 1445    | 889          | 1099      | 1404     |
> > | #Tasks as best method  | 8         | 4       | 1            | 0         | 5        |
> >
> > PhysFlow achieves strong performance, outperforming both FrameFlow and RFDiffusion in multiple tasks. Notably, PhysFlow achieves 1404 successes, substantially higher than FrameFlow (1099) and RFDiffusion (889). While Proteina obtains the highest overall numbers, it is trained on a significantly larger dataset, making the comparison not directly fair.
> >
> > Furthermore, PhysFlow can be naturally extended to sequence design. We can formulate a coupled sequence–structure noising process that extends the current unfolding-based noising mechanism to include sequence perturbations. The multimodal network can then jointly learn the reverse process for both structure and sequence generation.
> >
> > [1] De novo design of protein structure and function with RFdiffusion, Nature
> >
> >
> > > Many modern protein structure generation generate not only the backbone structure but all-atom structures including sequences.
> >
> > Yes, PhysFlow can be naturally extended to sequence and all-atom design. Specifically, we can formulate a coupled sequence–structure noising process that extends our current unfolding-based noising mechanism to include sequence perturbations. The multimodal network can then jointly learn the reverse dynamics for both structure and sequence generation, enabling coherent co-evolution of geometry and amino acid identity.
> >
> > Furthermore, a conditional refinement model can be trained alongside PhysFlow to place side-chain residues conditioned on the local rotational frame and the $C_{\alpha}$ coordinates, thereby extending the framework to full-atom modeling.
> >
> > Thank you very much for your constructive feedback! We hope we've sufficiently addressed your concerns, and if so, would appreciate a revised score.

---

### Official Review · Reviewer_8kS8 · 2025-11-01

**Soundness:** 2
**Presentation:** 2
**Contribution:** 2
**Rating:** 2
**Confidence:** 5

**Summary:**

PhysFlow introduces a physics-inspired non-linear noising process that targets the process of unfolding through secondary structures to avoid clashes. The central idea is to mimic the folding process during de novo design. PhysFlow leverages SE(3) flow matching along with sequence conditioning for both uncodnitional backbone design and forward sequence conditioned protein folding.

**Strengths:**

- Novel flow matching procedure. Clearly motivated and defined. The change from gaussian prior to leveraging Riemanian flow matching to go from linear sheet to de novo protein is very interesting.
- This framework allows for both structure generation and sequence-conditioned folding using SE3-equivariant architectures and a unified backbone/sequence conditioning, which enables accurate folding performance compared to some baseline models.

**Weaknesses:**

While interesting it seems to be an overly complex framework with little empirical benefits. PhysFlow seems to suffer from the  structural bias of early AlphaFold2-architecture-based backbone design methods as well. The motivation seems

- No analysis of the claim of linear flow models prone to steric clashes. Further the method cited RNA-Frameflow does not make this claim either.
- State-of the-art claims "PhysFlow generates the most designable sample" yet table 2 shows prior work with better designability, diversity, and novelty. Furthermore FoldFlow (OT) achieves 97.2% designability from lengths 50-250 as that matches its training length distribution from Geffner et al. Similarly Proteina is < 99% in their reporting. Overall it does not seem like a fair comparison as methods are compared outside of their training data length.
- Like FoldFlow, PhysFlow has significant alpha helix biased which is known to inflate designability.
- Folding benchmark not the same evaluation set as MultiFlow and FoldFlow2? Is there a reason a different evaluation was used? Could very well be a fair comparison but it is odd that FoldFlow-2 goes from ~3.2 to > 13 RMSD.
- No motif scaffolding when refereced as part of the motivation compared to prior backbone generative models, which likely may benefit from having a more physics grounded denoising process to place the motif correctly.

**Questions:**

- How long was the model trained for (time/steps)?
- What is the value of preserving physics during the generative process if only the final samples are evaluated and its backbone only compared to recent all-atom models?
- How does PhysFlow perform when trained on AFDB like recent state-of-the-art methods, Genie2 and Proteina? Also how long would it take to process AFDB to train PhysFlow to compared to those AFDB-using methods?
- How do the model size and inference speed compare to prior methods?
- Is there any reason why the physics-informed flow matching could not be used on cartesian CA-based models? Or is the SO(3) and angular factorizations a requirement?

---

> ### Author Response · Authors · 2025-11-20
> **Rebuttal**
>
> Many thanks for your constructive feedback. We address all your comments below.
>
> > No analysis of the claim of linear flow models prone to steric clashes.
>
> Thank you for the opportunity to clarify this. We have quantified the average number of residue–residue steric collisions (defined as pairwise residue distances $< d(x_i,x_j)$(nm)) observed across the forward trajectories of proteins. This analysis was performed for Cartesian linear diffusion/flow baselines as well as for variants of our PhysFlow method with different weightings $(k_1, k_2)$ as shown in Fig. 5 in the manuscript. We observe that the number of collisions decreases as the weight on Coulombic repulsion ($k_2$) increases within the PhysFlow framework, demonstrating that PhysFlow can effectively mitigate steric clashes compared to linear flow baselines.
>
> > State-of the-art claims...
>
> Thank you for the opportunity to clarify this aspect.
>
> ### Fairness of Comparisons
> We followed the evaluation protocol described in FoldFlow [1], which is also used by  baseline methods, including FrameDiff [2], FoldFlow-2 [3], and FrameFlow [4]. Specifically, FoldFlow (OT/CFM) was trained on monomer datasets from the PDB with sequence lengths ranging from 60 to 512 residues (see Appendix I.5 in [1]). Hence, sequences of length 300 fall well within the training distribution. Moreover, their evaluation explicitly included protein lengths in {100, 150, 200, 250, 300} (see Section 5.2 in [1]). We use the Proteina [5] model, which in their original work is also used to generate and evaluate longer protein sequences. RFDiffusion [6] is trained on the same data used for RoseTTAFold [7] and is additionally finetuned on sequences up to length 384. Chroma [8] is trained on sequences ranging from 50 to 1000 residues, with evaluation performed across the same range.
>
> Because these methods differ in training datasets and sequence-length cutoffs, we have used a dagger notation to clearly indicate cases where a method was not trained on exactly the same sequence lengths. Among the baselines, Genie is the only method trained up to 256 residues; we mark this with a dagger as well. We include Genie in our comparison for completeness, following prior baseline works [1,3,4]. To ensure a fair and consistent comparison, we evaluate all methods using the sequence lengths employed in the original protocols.
>
> In response to your suggestion, we have additionally evaluated proteins with lengths in the range 50–250 residues. We also compared our method to RFDiffusion without pretraining, as described in [2]. The results for designability, novelty, and diversity are summarized in table below,
>
> | Method                       | Designability | Novelty (TM-Frac) | Diversity (Cluster) |
> |------------------------------|---------------|--------------------|-----------------------|
> | RFDiffusion*                 | 0.64          | 0.40               | 0.45                  |
> | Proteina*                    | 0.98          | 0.38               | 0.30                  |
> | RFDiffusion (no-pretraining) | 0.24      | 0.36               | 0.31                  |
> | FrameDiff                    | 0.64          | 0.52               | 0.38                  |
> | FrameFlow                    | 0.88          | 0.47               | 0.51                  |
> | FoldFlow                     | 0.96          | 0.44               | 0.38                  |
> | PhysFlow                     | 0.97          | 0.44               | 0.43                  |
>
> We observe that RFDiffusion can generate designable sequences only when initialized with pre-trained weights, also evidenced by [2]. By contrast, PhysFlow achieves consistently high designability while maintaining competitive novelty and diversity across this broader range of sequence lengths. These results highlight the robustness of PhysFlow in generating structurally and functionally plausible proteins without relying on pretraining.
>
> [1] SE(3)-Stochastic Flow Matching for Protein Backbone Generation, ICLR 2024
>
> [2] SE(3) diffusion model with application to protein backbone generation, ICML 2023
>
> [3] Sequence-Augmented SE(3)-Flow Matching For Conditional Protein Backbone Generation, NeurIPS 2024
>
> [4] Fast protein backbone generation with SE(3) flow matching, preprint
>
> [5] Proteina: Scaling Flow-based Protein Structure Generative Models, ICLR 2025
>
> [6] De novo design of protein structure and function with RFdiffusion, Nature
>
> [7] Accurate prediction of protein structures and interactions using a three-track neural network, Science
>
> [8] Illuminating protein space with a programmable generative model, Nature
>
>
> > How long was the model trained for (time/steps)?
>
> The model was trained for 500k steps for 4 days on 1 H200 NVIDIA GPU.

---

> > ### Author Response · Authors · 2025-11-20
> > **Rebuttal**
> >
> > > Folding benchmark not the same evaluation set as MultiFlow and FoldFlow2?...
> >
> > In Table 3, our goal is to evaluate sequence-conditioned co-design approaches for monomer folding. To ensure a controlled and meaningful comparison, we compare against co-design models like MultiFlow and FoldFlow-2 using a curated training and testing dataset of monomers obtained from the PDB (see Appendix B.3). Both baseline methods are trained on the same training set and evaluated on the same test set, without any fine-tuning or additional datasets. This approach ensures a fair and consistent comparison across all methods.
> >
> > > No motif scaffolding....
> > ### Motif-Scaffolding Results
> >
> > Based on the your remark we have conducted a motif scaffolding experiment based on RFDiffusion benchmark [1]. To enable PhysFlow to perform motif-scaffolding, we add two additional features to our model:
> >
> > - The motif structure remains fixed (i.e., not unfolded), while only the remaining positions—determined by the motif mask—are unfolded during the forward simulation.
> >
> > - The fixed motif coordinates are also provided as conditional inputs to the model, supplying structural context throughout the denoising process.
> >
> > Here are the results on it compared to the baselines
> >
> > | Task Name              | Proteina† | Genie2† | RFDiffusion† | FrameFlow | PhysFlow |
> > |------------------------|-----------|---------|--------------|-----------|----------|
> > | #Successes total       | 2094      | 1445    | 889          | 1099      | 1404     |
> > | #Tasks as best method  | 8         | 4       | 1            | 0         | 5        |
> >
> >
> > PhysFlow achieves strong performance, outperforming both FrameFlow and RFDiffusion in multiple tasks. Notably, PhysFlow achieves 1404 successes, substantially higher than FrameFlow (1099) and RFDiffusion (889). While Proteina obtains the highest overall numbers, it is trained on a significantly larger dataset, making the comparison not directly fair.
> >
> > [1] De novo design of protein structure and function with RFdiffusion, Nature
> >
> > > What is the value of preserving physics during the generative process if only the final samples are evaluated and its backbone only compared to recent all-atom models?
> >
> > ### Value of Preserving Physics
> >
> > Thank you for the insightful question. While our primary evaluations focus on the generated backbone structures, preserving physical consistency throughout the generative process offers several important benefits:
> >
> > - **Inductive bias**: Enforcing physically consistent transformations during generation provides a meaningful inductive bias that guides the model toward realistic and energetically plausible conformations, improving sample quality and stability.
> >
> > - **Beyond static generation**: A physics-consistent formulation lays the foundation for modeling conformational dynamics—moving beyond static protein generation to capture dynamic processes such as folding pathways, transition states, and fold-switching behavior [1,2,3]. This opens promising directions for integrating generative models with protein physical simulations in future work.
> >
> >
> > [1] Scalable emulation of protein equilibrium ensembles with generative deep learning, Science
> >
> > [2] Simultaneous Modeling of Protein Conformation and Dynamics via Autoregression, NeurIPS 2025
> >
> > [3] Consistent Sampling and Simulation: Molecular Dynamics with Energy-Based Diffusion Models, NeurIPS 2025
> >
> > > How does PhysFlow perform when trained on AFDB like recent state-of-the-art methods, Genie2 and Proteina?...
> >
> > Thank you for the great question. The AFDB consists of approximately 21 million monomer structures following the filtering procedure used in FrameDiff and Proteina. Extending PhysFlow to AFDB is conceptually straightforward and would involve two main steps:
> >
> > -  Simulating the forward unfolding process, which can be efficiently parallelized across multiple CPUs.
> >
> > - Training PhysFlow on the simulated data.
> >
> > However, it is important to note that training on a dataset of this scale requires substantial computational resources, including large-scale GPU clusters and extended training time, which are not readily available in many research labs. While PhysFlow can, in principle, be scaled to AFDB, this practical limitation currently restricts large-scale experiments compared to smaller curated datasets.

---

> > > ### Author Response · Authors · 2025-11-20
> > > **Rebuttal**
> > >
> > > > Is there any reason why the physics-informed flow matching could not be used on cartesian CA-based models? Or is the SO(3) and angular factorizations a requirement?
> > >
> > > Thank you for the great question. The angular space is used to define the unfolding process as it preserves the structural integrity of the protein chain and prevents self-collisions. Operating in angular coordinates inherently enforces idealized bond lengths and local connectivity between residues—constraints that are not guaranteed when the flow is defined directly in Cartesian space. However, since the angular representation alone cannot fully capture inter-residue collisions, it is intermittently converted to Cartesian coordinates to evaluate the repulsion energy, which forms a component of our augmented potential. For completeness, we also describe an analogous flow in Cartesian space in Appendix C.2, though this formulation does not strictly maintain the bond-length constraints enforced in angular space.
> > >
> > >
> > > > How do the model size and inference speed compare to prior methods?
> > >
> > > We compare the model size and inference speed of PhysFlow against prior methods in the table below. Inference time is measured for generating a protein structure of length 100 on a GPU over 100 timesteps of the reverse diffusion process. Note that the large parameter count (650M) corresponds to the pretrained ESM language model used in FoldFlow-2 and PhysFlow.
> > >
> > > | Method       | Model size | Trainable params | Inference time (sec.) |
> > > |--------------|------------|------------------|-------------------------|
> > > | FrameDiff    | ~20M       | ~18M             | ~4.6                    |
> > > | FoldFlow     | ~20M       | ~19M             | ~4.7                    |
> > > | FoldFlow-2   | ~672M      | ~21M             | ~5.1                    |
> > > | PhysFlow | ~670M      | ~20M             | ~5.1                    |
> > >
> > > As shown, PhysFlow achieves a comparable inference speed to previous methods while maintaining a similar number of trainable parameters. This efficiency is notable given the inclusion of the large pretrained language model, which performs substantial computations.
> > >
> > > We are grateful for your thoughtful comments and suggestions, which have allowed us to emphasize some salient aspects, and shed light on subtle facets of the proposed method. We hope that your concerns have been sufficiently addressed, and if so, you would consider raising your score.

---

### Meta-Review · Area_Chair_Bbjm · 2026-01-21

**Summary:**

Almost all reviewers raise concern about the unjustified additional complexity of the proposed second-order Hamiltonian framework (Reviewer 8kS8, uDYi, 2ycY) and concerns about the empirical results. For example, reviewer 8kS8 and uDYi concerns about the bias toward Alpha-Helices. Reviewer 2ycY questioned about the justification for second-order velocity field, and reviewer uDYi questions about the fairness in comparison with baselines. There are other questions about additional experiments such as Motif Scaffolding (reviewer 8kS8 and 2bsB).

**Reviewer Concerns:**

The authors successfully addressed some concerns by providing empirical results such as quantitative results on Steric Clashes and a new motif scaffolding experiments. The question regarding the justification of the additional complexity and the bias of the dataset is not fully addressed.

**Reviewer Scores:**

The scores of the reviewers will likely to remain negative as reviewers expressed continuous concerns regarding the experiment results.

---

### Decision · Program_Chairs · 2026-01-26

Reject